# A negative feedback loop is critical for recovery of RpoS after stress in *Escherichia coli*

Sophie Bouillet [ID], Issam Hamdallah, Nadim Majdalani [ID], Arti Tripathi[¤], Susan Gottesman [ID]*

Laboratory of Molecular Biology, Center for Cancer Research, NCI, NIH, Bethesda, Maryland, United States of America

¤ Current address: Catalent Pharma Solutions, Baltimore, Maryland, United States of America
* Gottesms@mail.nih.gov

**Data Availability Statement:** All relevant data are within the manuscript and its Supporting information files.

## Abstract

RpoS is an alternative sigma factor needed for the induction of the general stress response in many gammaproteobacteria. Tight regulation of RpoS levels and activity is required for bacterial growth and survival under stress. In *Escherichia coli*, various stresses lead to higher levels of RpoS due to increased translation and decreased degradation. During non-stress conditions, RpoS is unstable, because the adaptor protein RssB delivers RpoS to the ClpXP protease. RpoS degradation is prevented during stress by the sequestration of RssB by anti-adaptors, each of which is induced in response to specific stresses. Here, we examined how the stabilization of RpoS is reversed during recovery of the cell from stress. We found that RpoS degradation quickly resumes after recovery from phosphate starvation, carbon starvation, and when transitioning from stationary phase back to exponential phase. This process is in part mediated by the anti-adaptor IraP, known to promote RpoS stabilization during phosphate starvation via the sequestration of adaptor RssB. The rapid recovery from phosphate starvation is dependent upon a feedback loop in which RpoS transcription of *rssB*, encoding the adaptor protein, plays a critical role. Crl, an activator of RpoS that specifically binds to and stabilizes the complex between the RNA polymerase and RpoS, is also required for the feedback loop to function efficiently, highlighting a critical role for Crl in restoring RpoS basal levels.

## Author summary

In their native environments, bacteria are exposed to constant changes in nutrient availability, as well as other biotic and abiotic stressors. To adjust to these changes, bacteria must rewire gene expression to adapt to or avoid stress-induced damage. A key player in the global response to general stresses is the alternative sigma factor RpoS, a promoter specificity -determining subunit of RNA polymerase. RpoS levels increase with stress, due to increased translation and stabilization of the otherwise unstable RpoS protein. Here, we examine how the cell restores homeostasis after the stress has passed. We show that a negative feedback loop in which RpoS regulates the

**Funding:** This work and all authors (SB, IH, NM, AT and SG) were supported by the intramural research program of the Center for Cancer Research, NCI, NIH. The funders had no role in study design, data collection and analysis, decision to publish, or preparation of the manuscript.

**Competing interests:** The authors have declared that no competing interests exist.

transcription of an adaptor for proteolysis poises the cell to rapidly resume RpoS degradation upon the exit from stress.

## Introduction

Bacteria have evolved to adapt to various stressful conditions. One way in which bacteria respond to these stresses is by the activation of alternative sigma factors, promoter recognition subunits of RNA polymerase that activate the transcription of specific sets of genes. The sigma factor RpoS is required for the transcription of general stress response genes in most γ-proteobacteria. In *E. coli*, RpoS promotes, directly or indirectly, the transcription of more than 500 genes that help cells respond and adapt to various stresses, including stationary phase, osmotic and heat shock, pH variation, and starvation for various nutrients [1–4]. RpoS is regulated at multiple levels, including its transcription and translation, stability of the protein, and modulation of its activity (reviewed in [5]).

RpoS proteolysis is strictly controlled to maintain homeostasis in cells through a unique degradation pathway mediated by the adaptor RssB. The adaptor binds to RpoS during rapid growth in the absence of stress, delivering it to the ClpXP protease. ClpXP, one of five ATP-dependent cytoplasmic proteases in *E. coli*, is composed of the chaperone ClpX, a member of the AAA+ family of proteins, and ClpP, forming the proteolytic component. While some substrates are directly recognized by ClpXP, others require an adaptor such as RssB. RssB is an orphan response regulator composed of a phosphorylatable receiver domain and an inactive PP2C-type phosphatase domain [6,7]. Mutating D58, the site of RssB phosphorylation, to an alanine or a proline reduces rather than inactivates RssB function, slowing RpoS degradation *in vivo* and *in vitro*. A mutation in D58 also does not abolish the ability of cells to stabilize RpoS in response to stresses, demonstrating that the stabilization is independent of RssB phosphorylation [8,9].

These observations led to the identification of anti-adaptors as the mediators of most stress-induced RpoS stabilization. Anti-adaptors interfere with the ability of RssB to bind to RpoS by competing with RpoS for RssB interaction, thus preventing the degradation of RpoS via the ClpXP complex. Three different anti-adaptors encoded in *E. coli* K-12 have been studied thus far; each is synthesized under different stress condition and stabilizes RpoS when produced. These three anti-adaptors are unrelated in sequence and structure, and each binds differently to RssB [9]. IraD responds to DNA damage and UV stress, IraM to magnesium starvation and low pH, and IraP to phosphate starvation [10–14]. IraP transcription is induced by a direct effect of ppGpp on its promoter during phosphate starvation [10]. Induction of each of these, by stabilizing RpoS, leads to a significant increase in RpoS level and activity.

Once made, RpoS must compete with other sigma factors for core RNA polymerase, and this process is also subject to regulation. RpoD activity is lowered through the actions of the anti-sigma factor Rsd and of 6S RNA. Both of these factors contribute to the successful competition of RpoS for RNAP binding [15]. The activity of RpoS is directly aided by the sigma factor activator Crl, widely present in *E. coli* and other Enterobacterales species [16]. Unlike canonical transcriptional regulators, this 16-kDa protein does not bind to DNA at promoters but rather binds and specifically stabilizes the RNA polymerase-RpoS sigma factor holoenzyme. As highlighted in recent structures of the complex of RNAP-RpoS-Crl-promoter DNA, Crl plays a role in promoting the formation of the complex and increasing the transcription of the genes that are under the control of RpoS [17–21]. In the absence of Crl, transcription of some RpoS-controlled genes is diminished, primarily when RpoS levels are low [22–24]. The

promoter for *rssB*, encoding the adaptor RssB, was shown to be under RpoS and Crl control [25,26]. Feedback loops controlling RpoS levels and activity are also known and make the RpoS regulatory network a very complex and intricate system (reviewed in [5,27]).

We and others have studied how regulation of RpoS synthesis and proteolysis changes as growing cells encounter different stresses. However, the return to homeostasis when the stress is gone is equally important. RpoS is subject to multiple levels of regulation, much of it post-transcriptional. At least three small regulatory RNAs (sRNAs) positively regulate RpoS translation by pairing with a region in the 5' UTR, thus opening the mRNA hairpin in the *rpoS* transcript loop and exposing the ribosome binding site. Each of the well-characterized sRNAs are transcribed under different conditions [reviewed in [5]]. When the stress has passed, the transcription of these sRNAs should stop. The *rpoS*-activating sRNAs are used stoichiometrically, pairing with a target mRNA and then being degraded [28].

Loss of the sRNAs should allow RpoS translation to return to its pre-stress state within minutes after the inducing stress ends. However, what happens to the RpoS that has already been made? It is at this step that degradation of RpoS may be particularly important. It was previously shown that RpoS becomes stable upon glucose starvation, but rapidly becomes unstable after glucose is restored [8,29], suggesting the existence of a robust mechanism for reducing RpoS levels during recovery. How this occurs, and whether it is similar during recovery from different stress conditions is not known. We began this study to investigate the nature of RpoS recovery from phosphate starvation, during which RpoS stabilization depends on IraP. We focus here on how Crl contributes in a critical fashion to RpoS degradation during recovery from phosphate starvation, via an RpoS-dependent feedback loop, and how those findings extrapolate to recovery from other stress treatments.

## Results

### RpoS degradation quickly resumes after phosphate starvation

RpoS protein produced during exponential phase undergoes degradation via the adaptor RssB and the ATP-dependent protease ClpXP. During phosphate starvation, RpoS is stabilized by the sequestration of RssB by the anti-adaptor IraP [10–12]. To explore how RpoS recovers after the release from phosphate starvation, isogenic WT and Δ*iraP E. coli* strains were grown in MOPS minimal medium to mid-exponential phase ($OD_{600} \approx 0.3$), filtered and resuspended in starvation medium (MOPS without phosphate). After one hour of growth in the starvation medium, chloramphenicol was added to block further translation and RpoS levels were monitored in a Western blot (labeled in this paper as a chase experiment). RpoS was stable and this stabilization was dependent upon IraP, as shown previously (Fig 1A and 1B). Phosphate was added back to the culture, and levels of RpoS were measured as a function of time to quantify RpoS accumulation; in this experiment, changing levels reflect both new synthesis and degradation. RpoS levels, high after starvation, rapidly dropped, with an RpoS half-life of approximately 8.5 minutes (Figs 1C, 1D, and S1A). This suggests that the stabilization caused by IraP is rapidly lost once phosphate is reintroduced into the culture. To directly determine the rate of RpoS degradation during recovery, cultures were first starved for phosphate, phosphate was added back for two minutes, followed by addition of chloramphenicol to block translation and measure RpoS degradation (Fig 1A and 1C). A comparison of RpoS accumulation to RpoS levels during the chase shows that RpoS levels dropped to a similar extent for the first 5 minutes, consistent with the decrease in accumulation being due to rapid degradation, with a half-life of about 5.5 minutes compared to over 25 minutes during phosphate starvation (Fig 1D, compare purple triangles to blue triangles). At later times, RpoS disappeared more quickly in the

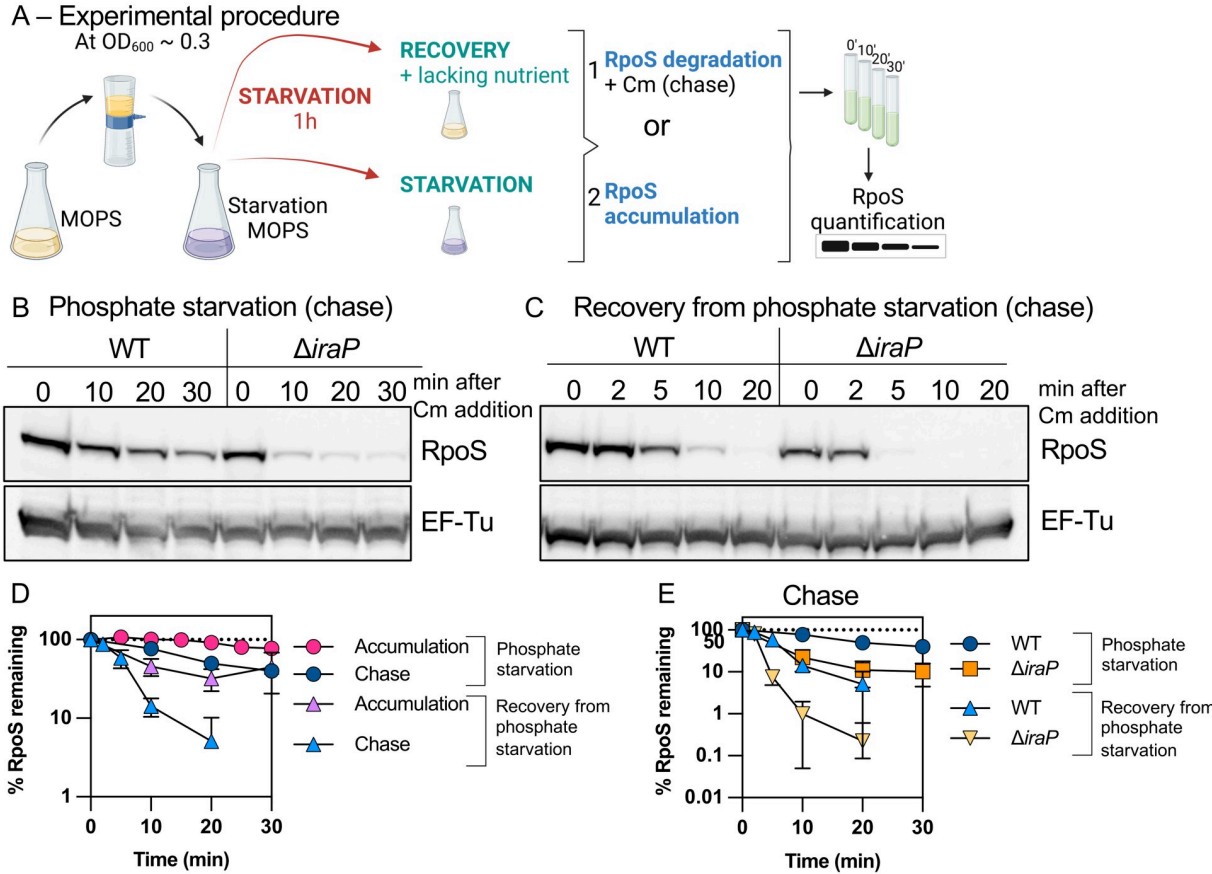

**Fig 1. RpoS degradation quickly resumes after phosphate starvation.** *A)* Experimental protocol. RpoS levels and degradation during and after phosphate starvation were determined as outlined here. Cells were grown in MOPS minimal medium containing 0.2% glucose and 2mM KPO$_4$ until mid-exponential phase (OD$_{600}$ ≈ 0.3). The medium was then filtered, and cells were resuspended in MOPS medium without phosphate and incubated for one hour (starvation). RpoS degradation during phosphate starvation was monitored by adding chloramphenicol and aliquots were taken right before (0 minute) and at appropriate time points afterwards (chase). RpoS recovery was measured in two ways. After one hour of starvation, PO$_4$ was restored to a portion of the culture (recovery), while another portion was further incubated under the starvation condition. Samples were taken from both cultures and analyzed either by monitoring RpoS levels with time (RpoS accumulation) or by determining RpoS half-life (RpoS degradation) by a chase after adding chloramphenicol 2 minutes after phosphate was restored and taking aliquots at different time points. After TCA precipitation, samples were loaded onto an SDS-PAGE gel and a Western Blot against RpoS, using EF-Tu as a loading control. Note that in this work, an antibiotic chase to block new protein synthesis is used to follow the half-life of proteins, rather than a radioactive or other label followed by a chase. One difference for such an antibiotic chase, as compared to a pulse-chase, will be that the protein followed during the chase is not necessarily newly synthesized, but both should provide similar estimates of degradation. This figure was created using clipart from BioRender.com. *B)* Western blot showing RpoS degradation (chase) during phosphate starvation in WT (MG1655) and Δ*iraP* (SB151) as described in Fig 1A. *C)* Western blot against RpoS and the loading control EF-Tu showing RpoS degradation (chase) during recovery from phosphate starvation in WT (MG1655) and Δ*iraP* (SB151) strains following the protocol as described in Fig 1A. *D)* RpoS levels in WT (MG1655) during and after phosphate starvation, with (chase) and without (accumulation) chloramphenicol, measuring RpoS degradation and overall RpoS levels, respectively (quantitation of WT data from experiments as shown in Fig 1B and 1C for the RpoS chase data and S1A Fig for the RpoS accumulation data). Shown are means with SD, n = 3. *E)* Effect of IraP on RpoS stabilization during and recovery after phosphate starvation. Quantification of RpoS degradation (chase) during and after phosphate starvation in WT (MG1655) and Δ*iraP* (SB151) strains (quantitation of Fig 1B and 1C; n > 3; note that data for WT is the same as in Fig 1D).

presence of chloramphenicol, consistent with new synthesis of RpoS contributing to the higher accumulated levels.

RpoS was even less stable in the absence of IraP during recovery (once phosphate was added back), with an RpoS half-life of around 2.7 minutes (Fig 1C and 1E). It is interesting to note that while IraP is the primary anti-adaptor leading to stabilization during phosphate

starvation, even in the absence of IraP, RpoS is more stable during starvation than during recovery (compare orange squares to orange triangles in Fig 1E). This suggests that an additional factor may be playing a role in either promoting RpoS stabilization during phosphate starvation or actively promoting RpoS degradation upon recovery from phosphate starvation.

The rapid decrease in RpoS levels (Fig 1) when phosphate is added back supports a model in which RpoS degradation, inhibited by IraP during starvation, resumes, and that rapid degradation is primarily responsible for the change in RpoS levels.

## RpoS degradation quickly resumes during recovery from glucose starvation and stationary phase

In order to compare the roles played by IraP in the recovery from phosphate starvation to recovery from other stresses, we measured degradation and recovery of RpoS levels in cells subjected to glucose starvation and in cells grown to stationary phase. Both conditions have previously been shown to lead to stable RpoS [12]. However, unlike phosphate starvation, no anti-adaptors have yet been identified as critical to sequester RssB during glucose starvation or stationary phase. IraD appears to contribute to increasing RpoS levels during the transition from late exponential to stationary phase but plays a minor role in stabilizing RpoS stability in this condition [30]. The low RpoS proteolysis after carbon starvation was found to correlate with reduced intracellular ATP levels, leading to slowed ClpXP-dependent degradation of RpoS; RpoS degradation was perturbed by low ATP levels much more than other ClpXP-dependent substrates [29].

We first confirmed that these stress treatments led to stabilization of RpoS, and then examined recovery from the stress. For glucose starvation, experiments were performed as described for phosphate starvation (Fig 1A). For stationary phase experiments, cells were grown overnight to reach stationary phase ($OD_{600} \approx 1.6$) and were either subjected to a chase experiment to measure RpoS stability (stabilization) or diluted back into fresh medium before chloramphenicol was added (recovery). Both conditions led to RpoS stabilization (S1B and S1C Fig), as previously observed [8,12]. After restoration of glucose or dilution of stationary phase cells into fresh media, RpoS degradation quickly resumed (Fig 2A and 2B). Interestingly, the rate of RpoS decay during recovery from phosphate starvation, stationary phase, and glucose starvation were very similar (Fig 2C). This suggests one or more common mechanisms leading to RpoS degradation during stress recovery may be at play.

The contribution of IraP to RpoS stability during and after these additional stresses was investigated. IraP was not needed for stabilization of RpoS during stationary phase (S1B Fig) and had only a modest effect on stability during glucose starvation (S1C Fig). However, it did play a role during recovery from these stresses (Figs 2D, S1B and S1C). In WT strains, RpoS half-lives measured soon after recovery began were 4–6 minutes for any of these stresses but were only 2 minutes in the absence of IraP (Fig 2D). This suggests that IraP expression is sufficient under these conditions to provide some protection to RpoS during recovery, independent of whether it is needed for the stress stabilization.

Given the somewhat unexpected role of IraP in recovery from glucose starvation and stationary phase, we further investigated whether the other two well-characterized anti-adaptors, IraD and IraM, also played roles in recovery, focusing on stationary phase. Single, double, and triple mutants of the three *ira* genes all still had stable RpoS during stationary phase (Figs 2E and S1D). During recovery, IraP was the primary determinant of whether recovery was as it was in WT cells (half-life of around 4–5 minutes) or more rapid (2', as seen in Fig 2F). Thus, under this condition (minimal medium growth to stationary phase) IraD and IraM did not play a significant role in either stabilization or recovery.

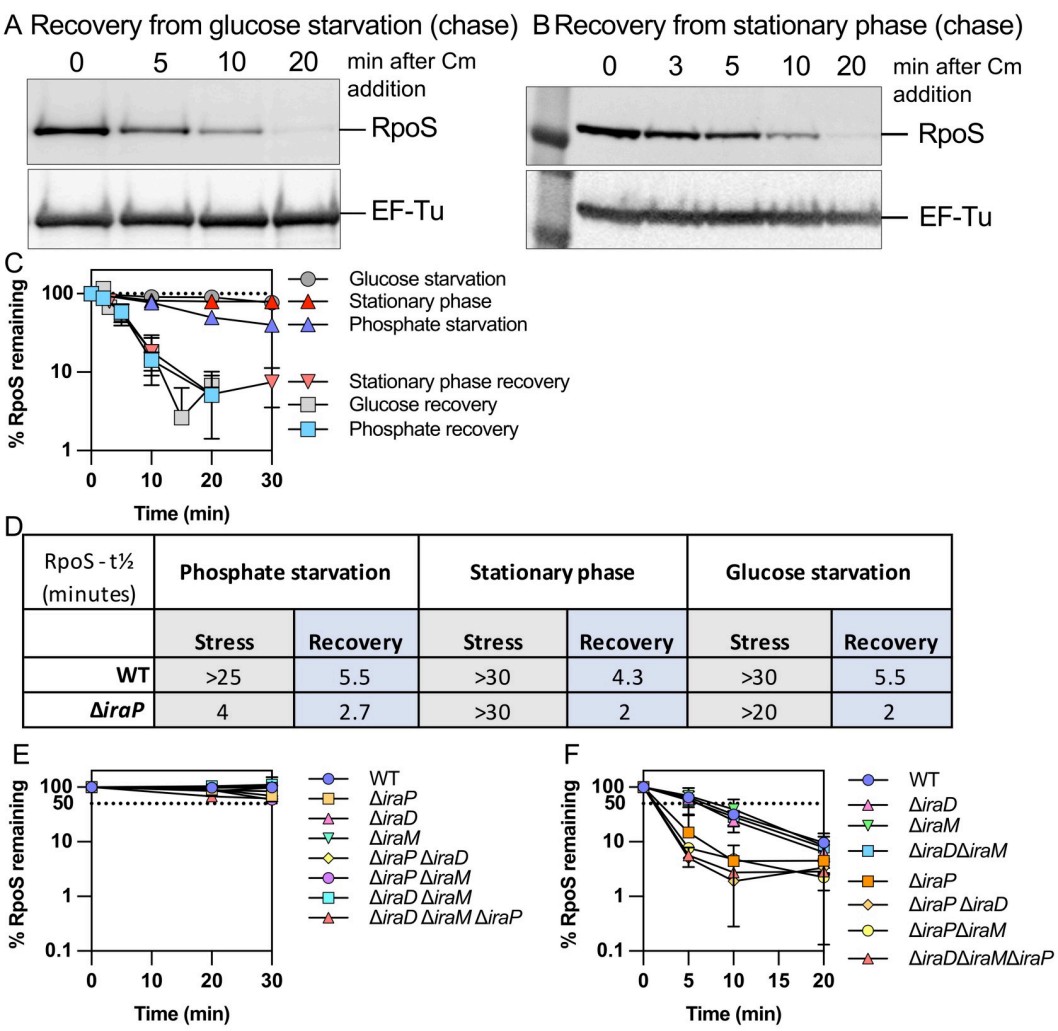

**Fig 2. RpoS levels recover quickly after glucose starvation and after stationary phase.** A) Western blot showing RpoS degradation after recovery from glucose starvation in MG1655 following the protocol described in Fig 1A. B) Western blot showing RpoS degradation after recovery from stationary phase in MG1655. Stationary phase cells were diluted back into fresh medium and chloramphenicol was added after 2 minutes. Samples were taken and treated as described for Fig 1. C) Quantification of RpoS degradation during and after phosphate starvation, glucose starvation and stationary phase (n > 3). Data collected from Western blots as in Figs 1B, 1C, 2A, 2B, S1B and S1C. D) RpoS half-lives in MG1655 and $\Delta iraP$ (SB151) strains during (stress) and after (recovery) phosphate starvation, glucose starvation and stationary phase as determined in experiments shown in Figs 1, 2 and S1. RpoS half-lives correspond to the time at which half of the $t_0$ RpoS protein disappears in a chase assay. E) Quantification of RpoS degradation during stationary phase (n > 3) in isogenic derivatives of MG1655 carrying various *ira* mutants. Strains assayed are listed in legend to S1 Fig. F) Quantification of RpoS degradation after stationary phase (n > 3) in isogenic derivatives of MG1655 carrying various *iraP* mutants. Strains assayed in 2E and 2F are listed in legend to S1 Fig.

## RpoS degradation after stress is an RpoS-dependent process

The rapid resumption of RpoS degradation during stress recovery is consistent with a need to restore RpoS homeostasis. It seemed possible that RpoS itself contributes to this process. To investigate whether RpoS is needed for its own degradation, we used a translational RpoS-LacZ fusion in *rpoS*+ and Δ*rpoS* strains. This fusion contains the 567-nucleotide leader upstream of the *rpoS* coding region, necessary for RpoS translational regulation, and extends through the first 786 nucleotides of the *rpoS* coding sequence, fused in frame to the coding sequence of

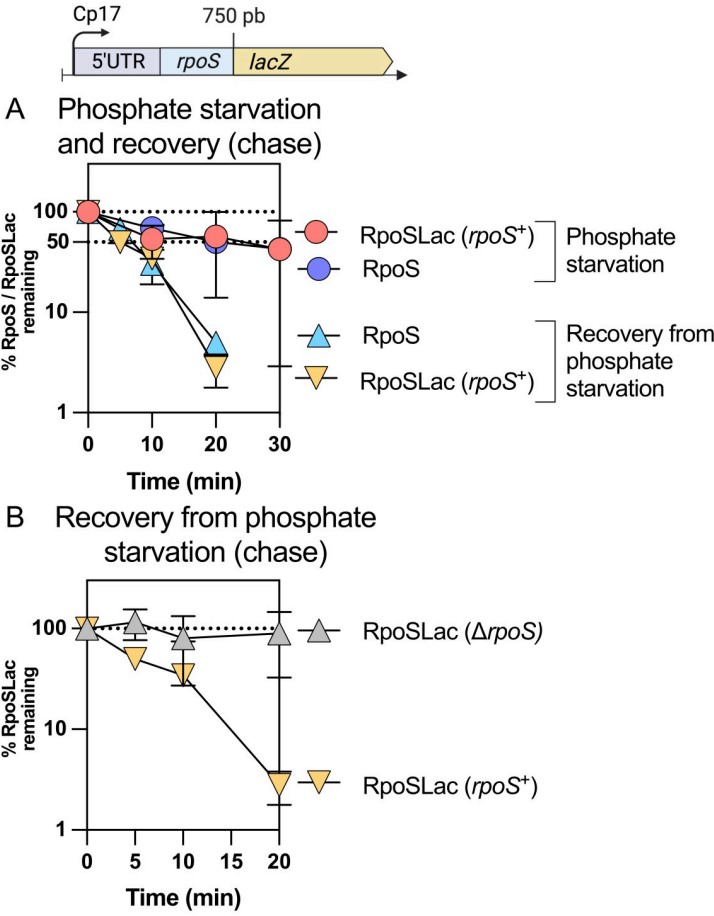

**Fig 3. RpoS-Lac recovery from phosphate starvation depends on RpoS.** A) RpoS and RpoS-Lac stabilization during phosphate starvation in the strain SG30013, a derivative of MG1655 containing both *rpoS*+ in the chromosome and the RpoS-Lac translational fusion at the *lacZ* site. The fusion contains the full 5' UTR of *rpoS* as well as 750 bp of the *rpoS* coding region, fused in frame to *lacZ* and expressed from a synthetic (Cp17) promoter. The protocol was as described in Fig 1A, a chase in which chloramphenicol was added to stop new translation. B) RpoS and RpoS-Lac degradation during recovery from phosphate starvation in the strains containing the RpoS-Lac translational fusion in the presence of RpoS (SG30013, yellow triangles), or in a strain deleted for RpoS (INH28, grey triangles); samples were taken as a function of time after chloramphenicol was added to stop translation.

*lacZ* (Fig 3). The RpoS portion of the fusion carries the residues required for RssB binding and for ClpXP-mediated RpoS degradation, including the critical K173 residue required for RssB recognition [31]. This fusion is therefore subject to the same translational and proteolytic regulation as the native RpoS. Transcription of the fusion is driven by a synthetic constitutive promoter, Cp17. We confirmed that the RpoS-LacZ fusion is inactive as a transcription factor by showing that it could not activate the transcription from the RpoS-dependent *gadB* promoter (S2A Fig). We first compared RpoS and RpoS-LacZ stabilization and degradation during phosphate starvation and recovery respectively in a strain carrying both a chromosomal copy of wild-type *rpoS* and the RpoS-LacZ translational fusion. Both RpoS and RpoS-LacZ were stable during phosphate starvation, and both RpoS and RpoS-LacZ levels dropped at similar rates during the recovery from phosphate starvation (half-life ~ 5.5 minutes, as seen in strain lacking RpoS-LacZ), consistent with a rapid restoration of degradation of both RpoS and RpoS-LacZ (Figs 3A, S2B, S2C and S2D). We then measured degradation of RpoS-LacZ during recovery

in a strain lacking native *rpoS*. After one hour of starvation, the RpoS-LacZ fusion protein remained stable and surprisingly, also remained stable when phosphate was added back (Figs 3B, S2B and S2D), with an RpoS-LacZ half-life > 20 minutes in the absence of RpoS, compared to 5.5 minutes in the presence of RpoS (Fig 3B). These results demonstrate that the presence of RpoS is required for its own degradation during recovery after stress, suggesting a role for one or more RpoS-dependent genes in the recovery.

## RssB transcription depends on RpoS

It was shown previously that *rssB* transcription is under RpoS control [25,26]. This suggests that RssB, necessary for RpoS degradation, might be one of the RpoS-dependent factors that becomes limiting during stress recovery in the absence of RpoS. To confirm the RpoS dependence of *rssB* transcription, we introduced a plasmid that contains an *rssB* translational reporter fusion into WT, Δ*rpoS* and Δ*rssB* strains. This fusion includes 1597 nucleotides upstream from the *rssB* coding gene as well as the first 24 bp of the *rssB* coding gene fused in frame to mCherry. The upstream region includes two promoters previously shown to drive *rssB* transcription, the P1 promoter, upstream of the *rssA* gene and the P2 promoter, in the intergenic region between *rssA* and *rssB* (Figs 4A and S3A) [25,26]. We confirmed that *rssB* transcription was dramatically lower in the absence of RpoS and increased during stationary phase, as seen previously (Fig 4A) [25,26]. In the absence of RssB, the *rssB* promoter reporter was not expressed at a significantly higher level, suggesting that wild-type RpoS levels may be sufficient to fully drive *rssB* transcription. We also measured RssB protein levels over the course of phosphate starvation in WT and Δ*rpoS* strains by western blot (Figs 4B, S3B and S3C). RssB levels doubled over one hour of phosphate starvation, while there was much less of an increase in RssB accumulation in a Δ*rpoS* strain. This confirms that the RpoS effect on *rssB* transcription is mirrored, during phosphate starvation, in effects on RssB protein accumulation. After phosphate was restored, the levels of RssB protein remained constant for at least one hour (Figs 4B, S3B and S3C).

As the anti-adaptor IraP is needed for stabilization during phosphate starvation and affects the kinetics of RpoS recovery from different stresses, we asked whether a change in the stoichiometry between IraP and RssB could be leading to the rapid degradation of RpoS. We compared the relative levels of RssB to those of IraP, C-terminally tagged with SPA, as we are unable to detect native untagged IraP with our anti-IraP antibody. *iraP* is transcribed from a ppGpp-dependent promoter that is induced early during phosphate starvation [10,32]. IraP-SPA protein levels increased during phosphate starvation, as expected, and did not decrease significantly when phosphate was added back (Figs 4C and S3D). Thus, RssB and IraP-SPA remained at similar proportions over the course of phosphate starvation and during recovery, suggesting that RpoS degradation during recovery is not linked to an increase of RssB relative to IraP. We also measured the stability of IraP-SPA using an antibiotic chase to block new synthesis during phosphate starvation and recovery from phosphate starvation (Figs 4D and S3E); IraP-SPA was stable in these conditions. As IraP appears to also play a role in RpoS degradation after glucose starvation and stationary phase, we quantified IraP-SPA levels in these conditions compared to those during phosphate starvation (S3D Fig). As seen previously, IraP-SPA levels increased during phosphate starvation and during stationary phase but did not increase significantly during glucose starvation [10]. Moreover, it is worth noting that IraP-SPA was present at high levels during stationary phase (S3D Fig), although IraP had no effect on RpoS stability in this condition (Figs 2D, 2E, S1B, S1D and S3D).

These results rule out some possible explanations for the rapid recovery from phosphate starvation. For instance, we see no evidence for a burst in RssB synthesis after addition of

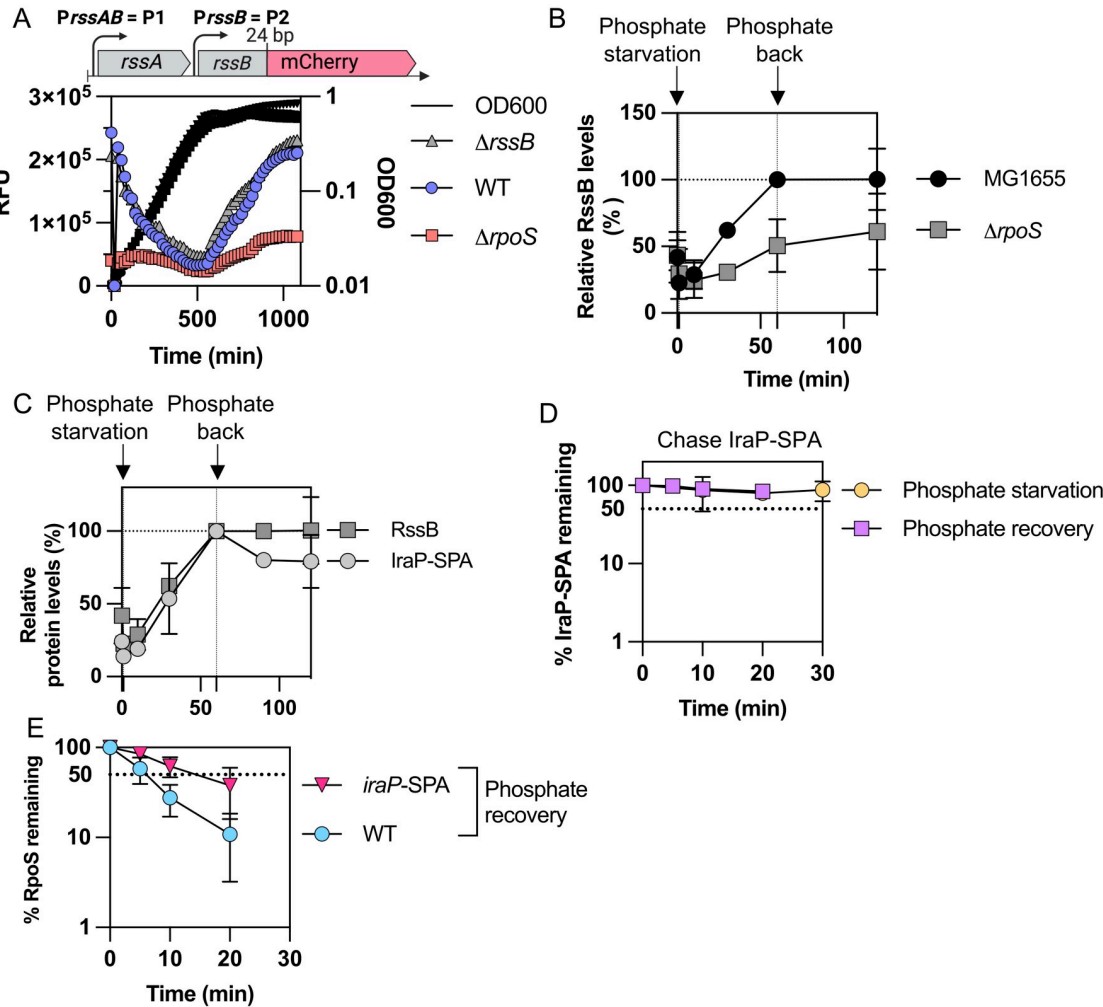

**Fig 4. RssB is produced during phosphate starvation dependent upon RpoS.** A) The upstream region of *rssB* that includes both *rssB* promoters P1 and P2 was translationally fused to mCherry contained on a pQE80L-derived vector (pSB37, S4A Fig). The fusion includes 24 bp of *rssB* (encoding 8 amino acids), fused in frame to mCherry. Expression of mCherry was measured during growth in MOPS glucose minimal medium at 37°C in WT, Δ*rpoS* (AB165) and Δ*rssB* (SB94) strains carrying the fusion on plasmid pSB37. The RFU (relative fluorescence units) were calculated by dividing the fluorescence values by the $OD_{600}$ at each time point. B) RssB levels measured in WT and Δ*rpoS* during and after phosphate starvation. The experiment was performed as described in Fig 1 with the exception that samples were probed for RssB in the Western blot (see S3 Fig for Western blots; n >2). RssB levels in MG1655 at the end of one hour of phosphate starvation were set at 100 and other samples are plotted compared to that. C) RssB and IraP-SPA tagged levels over the course of phosphate starvation and recovery. Strains MG1655 and SB212 (MG1655 containing *iraP* C-terminally tagged with SPA) were grown in MOPS minimal medium and subjected to phosphate starvation and recovery as described in Fig 1A. Samples were taken before starvation (0'), at 10, 30, and 60 minutes during starvation and after 30 and 60 minutes of recovery (corresponding to 90 and 120 minutes on the graph). Samples were subjected to a Western Blot using anti-Flag to detect the SPA C-terminal tag of IraP-SPA, and the bands were quantified. To compare RssB to IraP-SPA during starvation and recovery, levels of both proteins after 60' of starvation were set at 100% (N = 3). D) IraP-SPA chase after one hour of phosphate starvation and phosphate recovery, following a similar procedure as described in Fig 1A. Western Blot using anti-Flag antibody was performed to detect IraP-SPA (quantitation from Western Blot as shown in S3E Fig). E) RpoS chase during recovery from phosphate starvation in strains WT (MG1655) and SB212 containing *iraP*-SPA at the *iraP* locus (quantitation from Western blot as shown in S3F Fig).

phosphate (Fig 4C). Disappearance of IraP upon release of starvation could also explain the rapid recovery. No evidence of IraP degradation has been detected during *in vitro* anti-adaptor assays [9]. Evaluation of the fate of IraP in vivo is more difficult because thus far, we can only measure the tagged protein. There was no evidence of degradation of IraP-SPA (Figs 4D and

S3E), but the SPA tag did interfere with rapid recovery (Fig 4E). The longer stabilization caused by IraP-SPA during recovery does suggest that characteristics of the IraP protein contribute to rapid recovery.

## Crl mediates activation of various RpoS-dependent promoters, including the *rssB* promoters

As described in the introduction, Crl is a protein that helps stabilize the interaction of RpoS with core RNA polymerase, thereby increasing RpoS activity. A previous study had suggested a role for Crl in stationary phase, particularly at low RpoS levels, and demonstrated that Crl unexpectedly stimulated RpoS degradation via its role in promoting the expression of the RpoS-dependent *rssB* promoter [24]. Given the requirement for RpoS in rapid recovery from starvation, we investigated the importance of Crl in the feedback loop for *rssB* transcription and stress recovery.

First, the role of Crl was compared to that of the IraP anti-adaptor, both positive regulators of RpoS but with very different modes of action, on a set of plasmid-based mCherry transcriptional fusions for RpoS-dependent promoters (S4A Fig). It was previously reported that Crl would affect specific sets of RpoS-dependent promoters [22,33,34], or was mainly active when RpoS concentrations are low in the cell [34]. Our promoter set included P*gadB*, reported to be strongly Crl dependent, P*osmE*, reported to be relatively Crl-independent, as well as P*yodD* and P*osmY* promoters [22]. Isogenic WT, Δ*rpoS* and Δ*crl* strains transformed with the transcriptional fusions were grown and mCherry levels followed during growth and compared during stationary phase (S4B Fig). While the four fusions had different levels of activity (compare y axis values for each of the graphs in S4B Fig), all showed the expected RpoS dependence, with *yodD* the most purely RpoS dependent. Unexpectedly, the impact of Crl appeared similar, reducing activity to about 75% of the WT strain activity for all the promoters (S4D Fig; averaged data combining all four promoters shown in S4C Fig). Note that we had previously reported that *dsrB*, the gene transcribed divergently from *yodD*, was RpoS dependent [35,36]. Subsequently, we have determined that the fusion construct used in that work is in fact a *yodD-lacZ* fusion (see Materials and methods for more details).

We then investigated the impact of IraP and potential cooperation between Crl and IraP on RpoS activity, comparing the activity of the same plasmid-borne fusions in a Δ*iraP* or Δ*iraP* Δ*crl* strain to that in the WT strain. Deleting *iraP*, like deleting *crl*, resulted in a reduction of RpoS activity to approximately 75% of WT RpoS activity (S4B, S4C and S4D Fig). However, when both IraP and Crl were deleted, we observed a partial additive effect of Δ*crl* and Δ*iraP*, confirming that these two proteins likely play independent but parallel roles in RpoS regulation under these conditions (see averages in S4C Fig). Note that in these experiments, *iraP* expression was not induced by phosphate starvation but by entry into stationary phase. Therefore, even though we did not detect a role for IraP in the stabilization of RpoS in this condition (Figs 2D, 2E and S1B and S1D), IraP must be synthesized sufficiently to affect the level of RpoS activity (S4C and S4D Fig). Parallel experiments were carried out testing the effect of deleting *crl* on the *rssB*-mCherry transcriptional and translational fusion used in Fig 4. Crl affected *rssB* transcription similarly to its effects on other RpoS-dependent promoters (S4D and S4F Fig).

## The RpoS activator Crl is necessary for proper RpoS degradation after stress and acts in an RpoS- and RssB-dependent manner

The results above confirm the role of Crl in promoting the activity of RpoS. Given that we also see a critical role for RpoS-dependent transcription of *rssB* for recovery from phosphate starvation, we expected that the loss of Crl might perturb the recovery of RpoS degradation after

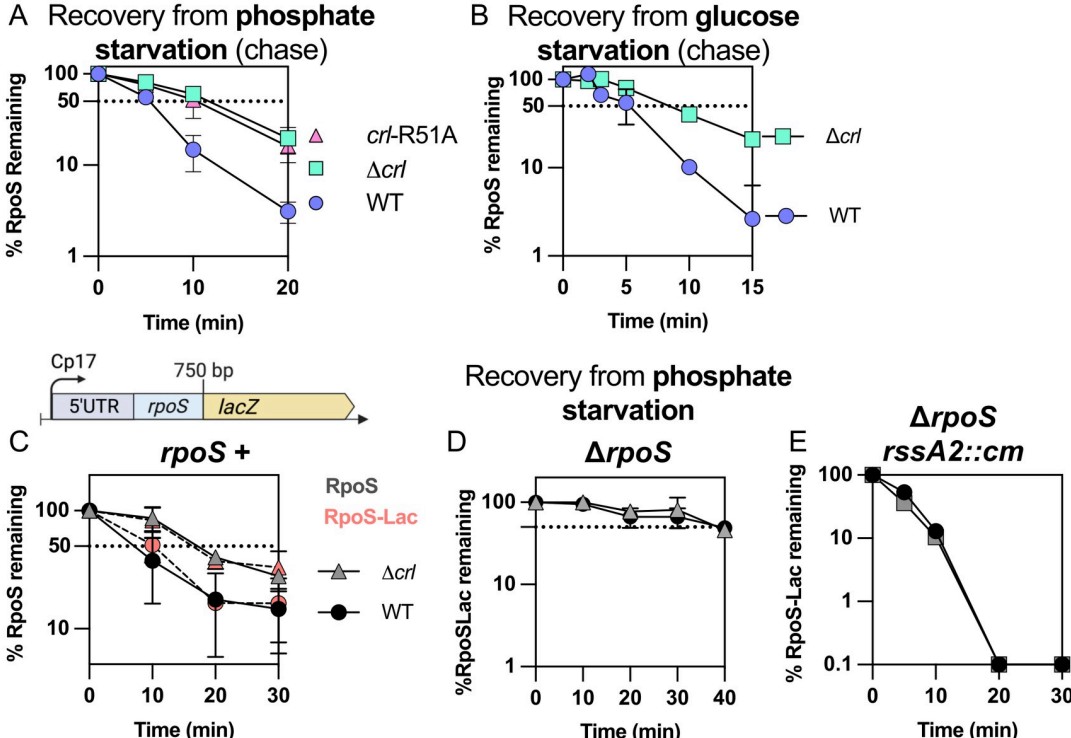

**Fig 5. Rapid RpoS recovery from phosphate and glucose starvation requires Crl.** A) RpoS degradation during recovery from phosphate starvation in the WT, Δ*crl* (SB147) and *crl*-R51A (SB148) strains (quantitation from Western Blot as shown in S5A Fig). Experiment was performed as described in Fig 1A. B) RpoS degradation during recovery from glucose starvation in the WT and Δ*crl* (SB147) strains (quantitation from Western Blot as shown in S5B Fig). Experiment was performed as described in Fig 1A. C) RpoS and RpoS-Lac accumulation experiments (no chloramphenicol added) during recovery from phosphate starvation in strains SG30013 (*crl*⁺) and SB180 (Δ*crl*), both containing the RpoS-Lac fusion; quantitation is from Western Blot as shown in S6C Fig. D) RpoS and RpoS-Lac accumulation experiments (no chloramphenicol added) during recovery from phosphate starvation in WT (*crl*⁺; circles) and Δ*crl* (triangles) strains containing the RpoS-Lac fusion but lacking *rpoS* (INH28 (*crl*⁺) and SB176 (Δ*crl*); quantitation is from Western Blot as shown in S6C Fig. E) RpoS and RpoS-Lac accumulation experiments during recovery from phosphate starvation in strains lacking *rpoS* and carrying *rssA2::cm*, which overproduces RssB, SB150 (*crl*⁺) and SB174 (Δ*crl*), both containing the RpoS-Lac; quantitation is from Western Blot as shown in S6C Fig.

release from stress. While Crl did not have a significant effect on RpoS stabilization during phosphate or glucose starvation (S5C and S5D Fig), it was necessary for rapid recovery. The absence of Crl doubled the RpoS half-life after recovery, to ~ 10 minutes, compared to ~ 5 minutes in a WT strain (Figs 5A, 5B, S5A and S5B, half-lives shown in S5E Fig).

Two approaches were used to confirm that Crl was exerting its influence on recovery via its effect on RpoS. First, recovery was examined in cells carrying an allele of Crl, Crl-R51A, a mutation at the RpoS binding interface, previously shown to be defective for RpoS interaction [19]. We confirmed the interaction of RpoS and Crl and loss of interaction with the Crl-R51A allele using a bacterial two-hybrid experiment (S6A Fig). In our assays, a chromosomal *crl*-R51A allele behaved like a *crl* deletion, slowing RpoS degradation during recovery from phosphate starvation (Figs 5A and S5A) and reducing activation of the *rssB* promoter (S6B Fig). Thus, the role of Crl for RpoS recovery from stress was abolished by a point mutation that abrogates the interaction with RpoS and loses the ability to stimulate expression of the RpoS-dependent *rssB* promoter, consistent with Crl acting in its previously described fashion.

If Crl is working via RpoS, the Crl effect on RpoS recovery should require RpoS. This was examined in strains expressing RpoS-LacZ in the presence and absence of native RpoS (Figs 5C, 5D and 5E, and associated Western blots in S6C and S6D Fig). The kinetics of recovery of RpoS-LacZ and RpoS were slowed to the same extent in a strain deleted of *crl* (Figs 5C and S6C). Since RpoS associated with core polymerase should be protected from RssB-dependent degradation [24], this result would suggest that any stimulation by Crl of the functional inter-action of RpoS with core polymerase, while clearly important for RpoS function, did not con-tribute significantly to recovery kinetics. As expected, *crl* mutants had no further effect when RpoS was absent (Figs 5D and S6C).

In a parallel set of experiments, we examined RpoS-LacZ recovery in a strain in which RssB was up-regulated by a transposon insertion in *rssA*, the gene upstream of *rssB*. This allele, *rssA2*::*cm*, leads to constitutive overproduction of RssB [26]. Overproduction of RssB led to a rapid loss of RpoS-LacZ after starvation ended, independent of RpoS (Figs 5E and S6C). More-over, *crl* had no further effect when *rssB* was under control of the *rssA2*::cm promoter and thus was no longer RpoS-dependent (Fig 5E). It is worth noting that the absence of RpoS led to a striking increase in RpoS-Lac levels even before phosphate starvation (S6E Fig), suggesting that a negative feedback loop regulates RpoS levels in the absence of stress. However, this RpoS-dependent feedback is overcome by increasing RssB levels with the *rssA2*::cm mutation (S6E Fig), consistent with our observation that overexpressing *rssB* is sufficient to bypass the requirement for RpoS for recovery.

## Crl is a critical part of a feedback loop allowing recovery from stress

Our results thus far support a feedback loop ensuring RpoS homeostasis via degradation. RpoS is negatively regulated by ClpXP degradation, dependent upon the RssB adaptor. Because RpoS, aided by Crl, promotes *rssB* transcription, whenever RpoS activity increases, so should the transcription of *rssB*. More RssB then poises the system for degradation of free RpoS. Dur-ing phosphate starvation, the increase in RssB is not sufficient to lead to RpoS degradation, at least in part because of IraP inhibition of RssB. Upon the restoration of phosphate, a switch, still to be defined, leads to the release of RssB and thus degradation of RpoS. In the absence of the feedback loop (i.e., in the absence of Crl or transcriptionally functional RpoS), there is not sufficient RssB to cause rapid RpoS degradation.

This model was further tested by examining the epistasis of mutations in *crl*, *rpoS* and *rssB* on expression of a chromosomal *rpoS*-mCherry translational reporter, that, like the RpoS-LacZ reporter, is subject to RssB-dependent degradation and is constitutively expressed from a syn-thetic promoter. This allows the direct measurement of mCherry fluorescence, and therefore RpoS levels, throughout growth, as previously described for the transcriptional fusions in Fig 4. As seen for RpoS-Lac quantification in S6E Fig, we observed a significant increase in RpoS-mCherry level in the absence of RpoS, compared to the *rpoS*+ parent (compare red to blue in Fig 6A), consistent with a negative feedback loop in which one or more RpoS-dependent func-tions reduce the levels of the fusion protein. One of these negative regulators is likely RssB, because the deletion of *rssB* also led to an increase in RpoS-mCherry levels (Fig 6A). Overpro-duction of RssB, as in the *rssA2* strain, reduced RpoS-mCherry (S7A Fig). However, because a deletion of *rpoS* increased levels of the fusion even more than loss of RssB (compare Δ*rssB* to Δ*rpoS* in Fig 6A), at least one other component of the RpoS regulon must participate in nega-tive feedback control of RpoS levels later in stationary phase.

As Crl was shown to play a key part in the RpoS feedback loop, we investigated the impact of a *crl* deletion on RpoS-mCherry levels. The deletion of *crl* or a *crl*-R51A mutation led to sim-ilar increases in RpoS-mCherry levels (Figs 6B and S7A), confirming a requirement for Crl in

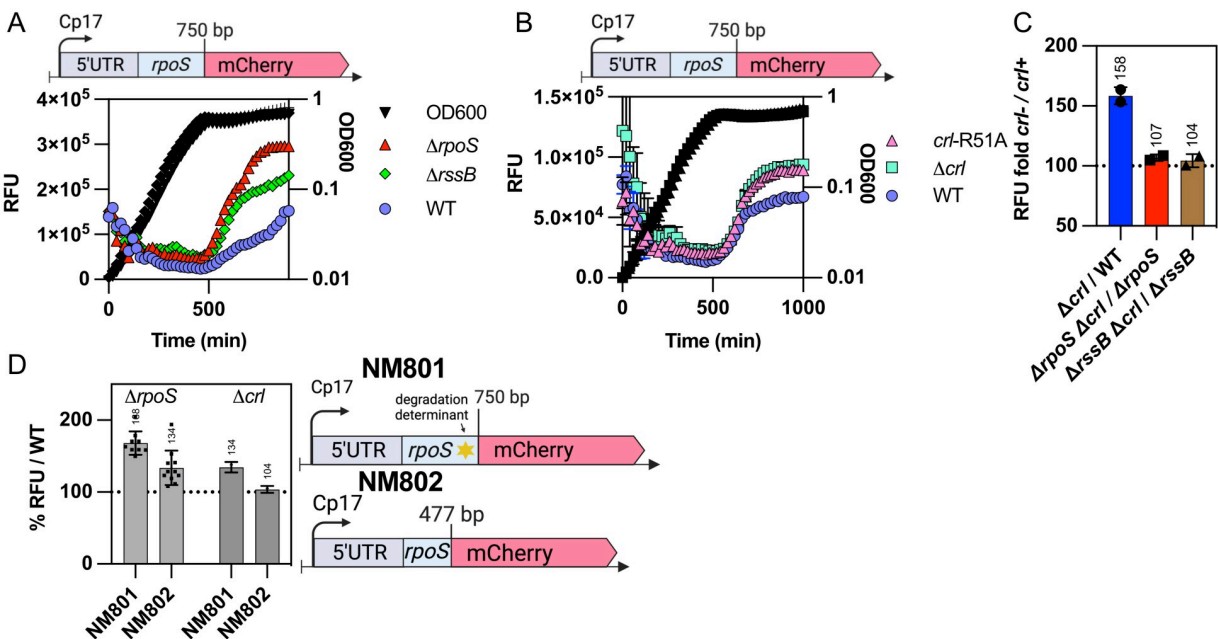

**Fig 6. Crl-dependence for RpoS-mCherry levels depends on RpoS.** A) The WT (NM801), Δ*rpoS::tet* (SB238) and Δ*rssB::tet* (SB225) strains containing the RpoS-mCherry translational fusion were grown and mCherry fluorescence measured over time as described in Fig 4A. B) The WT (NM801), *crl*-R51A (SB341) and Δ*crl::kan* (SB228) strains containing the RpoS-mCherry translational fusion were grown and mCherry fluorescence measured over time. C) The effect of the absence of Crl on RpoS-mCherry in the absence of RpoS or in the absence of RssB (high levels of RpoS) was determined by measuring mCherry fluorescence of the WT (NM801), Δ*rpoS::tet* (SB238), Δ*rssB::tet* (SB225), Δ*crl::kan* (SB228), Δ*crl::kan* Δ*rpoS::tet* (SB239) and Δ*crl::kan* Δ*rssB::tet* (SB283) strains, all containing the RpoS-mCherry fusion, over time as described in Fig 4A. RFU ratios between the strains containing the deletion of *crl* and the corresponding single Δ*rpoS* and Δ*rssB* mutant or WT were calculated from the data shown in S7B Fig (see S7C Fig for full graph). Ratios shown here are from stationary phase (t = 800 minutes). D) The negative feedback loops involved in RpoS regulation were compared in RpoS-mCherry fusions that are subject to both translational and proteolytic regulation (derivatives of NM801 containing RpoS750-mCherry, with 250 amino acids from RpoS, including the degradation determinant around amino acid K173), or fusions unable to be degraded (derivatives of NM802, containing RpoS477-mCherry, with 159 aa from RpoS). mCherry fluorescence was measured over time as described in Fig 4A and the RFU ratios at 800 minutes (during stationary phase) were calculated for *rpoS⁻* or *crl⁻* derivatives of each fusion compared to WT (from data shown in S7E Fig). RpoS-mCherry strains: WT (NM801), Δ*rpoS::tet* (SB238) and Δ*crl::kan* (SB228); RpoS477-mCherry strains: WT (NM802), Δ*rpoS::tet* (SB439) and Δ*crl::kan* (SB437).

this feedback loop and that the Crl effect is dependent upon its ability to bind to RpoS. We then examined whether the effect of Δ*crl* on RpoS-mCherry required RpoS or RssB. The RFU values of WT, Δ*crl*, Δ*rpoS*, Δ*rssB* strains and of the double mutant strains Δ*crl* Δ*rpoS and* Δ*crl* Δ*rssB* were determined and the ratio between *crl⁺* and Δ*crl* during stationary phase calculated (Fig 6C; graphs with growth are shown in S7B Fig and ratios across growth are shown in S7C Fig). As expected, the Crl effect on RpoS-mCherry was dependent on both RpoS and RssB (Figs 6C and S7C). The deletion of *crl* increased RpoS-mCherry levels to 158% compared to the *crl+* strain, but this difference was lost in the Δ*rpoS* or Δ*rssB* strains. Overproducing RssB from the RpoS-independent *rssA*2 promoter also eliminated the effect of Crl on RpoS-mCherry (S7D Fig).

We also examined the feedback loop in a strain expressing a shorter RpoS-mCherry fusion that lacks the RssB recognition site within RpoS and is thus not degradable (NM802, Figs 6D and S7E). While the increase in expression of the fusion in an *rpoS* mutant is not nearly as striking as seen with the degradable NM801 fusion, it is still significant (134% increase in RpoS-mCherry compared to WT). Since the two fusions carry the same synthetic promoter, this additional feedback effect is likely to be exerted on RpoS translation rather than

transcription. Intriguingly, a *crl* mutant in the presence of native RpoS had essentially no effect on the non-degradable RpoS-mCherry fusion, consistent with the loss of Crl effect in the *rssB* mutant.

The effect of Crl on RpoS activity was also evaluated, by measuring the activity of the *gadB* promoter (S7F and S7G Fig). In *rssB*⁺ cells, a *crl* mutant had very little effect during early exponential phase (ratio close to 100 (*crl*⁺) value, S7G Fig), but had somewhat lower relative RpoS activity later in exponential and stationary phase. In the absence of RssB, Crl had a strong effect on *gadB* expression early in exponential growth (brown triangles in S7G Fig below dotted line at 100, *crl*⁺ value). This result could suggest that the effect of Crl on RpoS activity may be hidden by degradation feedback when RssB is available and active, as it would be early in exponential phase. In the absence of degradation, the effect on RpoS activity becomes evident.

## RssB may be fully phosphorylated *in vivo* and its phosphorylation status does not impact RpoS degradation after stress

Our results thus far have implicated RpoS-dependent transcription of *rssB* as part of a critical feedback loop necessary for the rapid recovery from phosphate starvation, glucose starvation and stationary phase. Reducing RpoS activity modestly with a *crl* mutation (about 25% reduction compared to WT activity) was sufficient to block full recovery. However, this RpoS effect on RssB levels occurs during starvation, and does not provide an explanation for the rapid switch from stable RpoS to unstable RpoS upon the exit from starvation. Thus, we considered other factors known to affect RpoS availability and RssB-dependent degradation. One of these is the state of RssB phosphorylation. RssB is a phosphorylatable response regulator, phosphorylated on residue D58. It has been shown that the unphosphorylatable variants such as D58P or D58A exhibit a decreased affinity for RpoS *in vitro* and modestly increase RpoS stability *in vivo* [8,9]. However, the mutant form of RssB is still sensitive to the action of anti-adaptors such as IraP [9]. RssB is an orphan response regulator, meaning that it does not have a known associated histidine kinase, but rather appears to be phosphorylated by various phosphate donors *in vivo*, including the histidine kinase ArcB [37] and potentially other histidine kinases. We investigated the role of phosphorylation of RssB after phosphate starvation by creating isogenic strains in which the aspartate 58 residue in the chromosomal copy of *rssB* was replaced by alanine or proline to block formation of the phosphorylated state of RssB, or by glutamine to mimic a constitutive phosphorylated state. The expression of the P*gadB*-mCherry fusion was measured in these derivatives during growth (Fig 7A). Both the *rssB*-D58P and *rssB*-D58A mutations led to an increase in RpoS activity, although less than that seen in a Δ*rssB* mutant, consistent with previous results. The *rssB*-D58E variant was indistinguishable from WT RssB for this measure of RpoS activity, suggesting that RssB protein is likely to be mostly phosphorylated under these growth conditions.

Given these results, we investigated the effects of these mutations on RpoS degradation during recovery from phosphate starvation (Figs 7B and S8). Only minor differences were observed in RpoS stability regardless of the RssB variant used, suggesting that the phosphorylation state of RssB has negligible impact on the rapid recovery of RpoS degradation after stress is relieved. Therefore, we conclude that phosphorylation or dephosphorylation of RssB is not involved in the recovery from phosphate starvation.

## Additional known regulators of RpoS activity regulation are not involved in RpoS degradation after stress

Additional regulators of sigma factor activity are known in *E. coli*, including two that are under the control of RpoS. Rsd, an anti-RpoD factor, and 6S RNA, encoded by *ssrS*, both

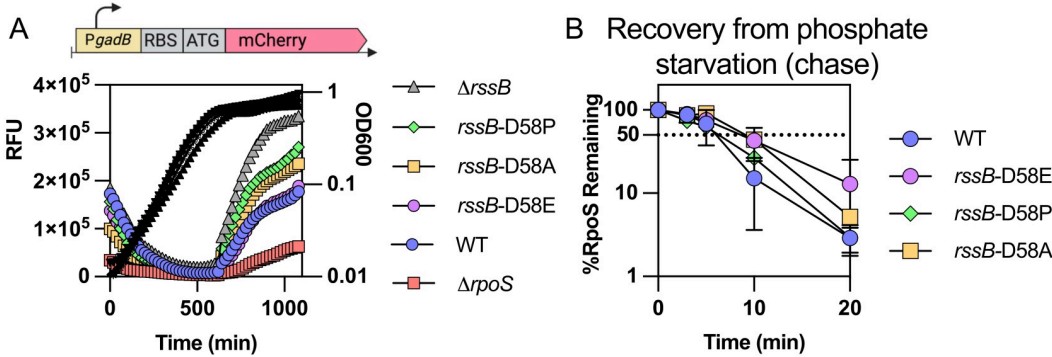

**Fig 7. The phosphorylation status of RssB does not impact RpoS recovery from phosphate starvation.** A) The unphosphorylatable RssB-D58 alleles D58P and D58A and the phosphomimic allele D58E were tested for RpoS activity by following the expression of the transcriptional fusion between *gadBp* and mCherry expressed on a vector (pSB23) in the strains MG1655, Δ*rpoS::tet* (AB165), Δ*rssB::tet* (SB94), *rssB*-D58E (SB192), *rssB*-D58A (SB190) and *rssB*-D58P (SB198). B) RpoS degradation after phosphate starvation of MG1655, *rssB*-D58E (SB192), *rssB*-D58A (SB190) and *rssB*-D58P (SB198). Shown are means with SD, n = 2. Example western blots are shown in S8 Fig.

inhibit RpoD activity, through different mechanisms, increasing the availability of core RNA polymerase for alternative sigma factor binding [38,39]. We tested the effect of strains lacking Rsd or 6S RNA on RpoS degradation during phosphate starvation and during recovery from starvation after chloramphenicol addition and did not observe any difference in RpoS half-lives in these strains (Figs 8 and S9). This result shows that these factors, despite affecting alternative sigma factor activity, are not involved in RpoS degradation after stress. In addition, since these factors should affect core polymerase availability for RpoS, the absence of an effect suggests that this is not a limiting factor for RpoS degradation during recovery, as also suggested by the similar rate of degradation of RpoS and RpoS-LacZ, the latter unable to activate core polymerase, seen in S3A Fig.

## Crl and IraP play parallel roles in regulating RpoS-dependent activities

Crl and IraP each act to positively regulate RpoS function, but by very different means, with Crl improving RpoS activity and IraP stabilizing RpoS. They have similar but somewhat

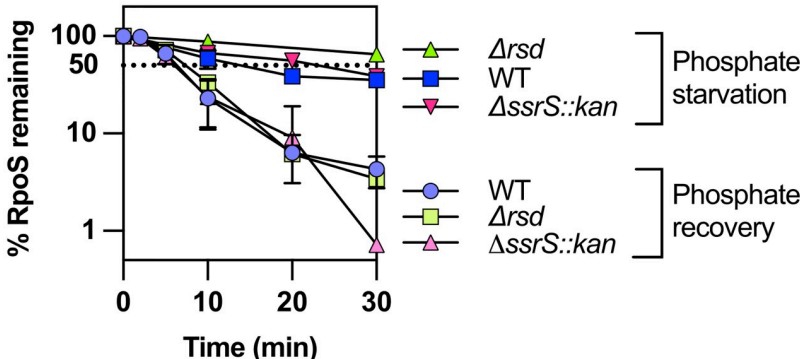

**Fig 8. RpoD activity regulators Rsd and 6S RNA do not impact RpoS degradation after phosphate starvation.** RpoS stability evaluated by adding chloramphenicol to stop translation after 1 hour of phosphate starvation and during recovery from phosphate starvation as described in Fig 1A in WT, Δ*rsd* (SB505) and Δ*ssrS::kan* (SB470) strains (quantitation from Western Blot as shown in S9A and S9B Fig).

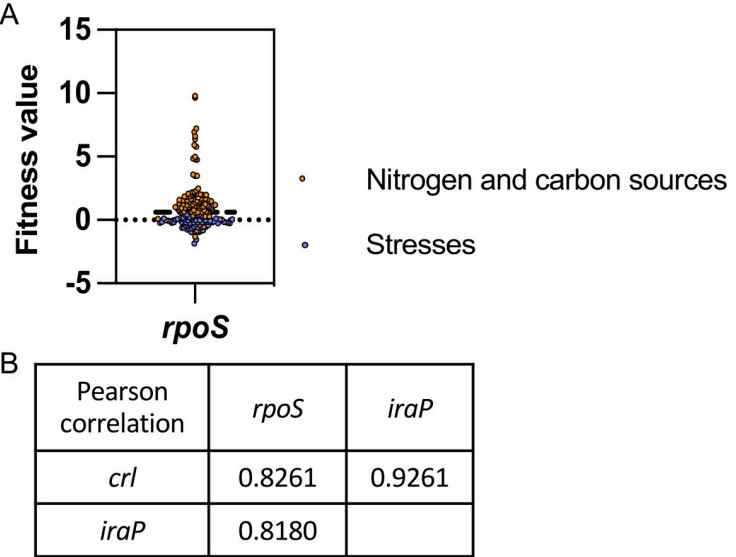

**Fig 9. Fitness analysis of *rpoS* gene.** A) Fitness values (log2 ratios) for an *rpoS* mutant from the fitness browser database (https://fit.genomics.lbl.gov/cgi-bin/myFrontPage.cgi). Positive values mean the gene is detrimental while negative values mean the gene is beneficial for fitness. B) Pearson correlation of fitness values for all conditions tested (168) of *rpoS*, *crl* and *iraP*.

additive effects on RpoS-dependent promoters (S4 Fig). We used the Fitness Browser database to further compare the roles of Crl and IraP. This website provides the relative growth fitness of transposon insertions throughout the *E. coli* genome under a variety of conditions, and provides a correlation in fitness behavior of inactivation of different genes [40]. Fitness data for insertions in *rpoS* clearly showed that loss of RpoS is beneficial for growth in most carbon and nitrogen sources, while slightly detrimental for some antibiotic stresses (Fig 9A). Insertions in *crl* and *iraP* (*yaiB* on this site) had strong correlations with *rpoS* (Pearson correlations of .8), and, most strikingly, a correlation with each other of better than 0.9 (Fig 9B). This high correlation is usually found for multiple subunits of a complex; however, we have seen no evidence of this for IraP and Crl and the additivity of mutants (S4 Fig) suggests parallel but independent activities. Other positive regulators of RpoS, including the other anti-adaptors and small RNAs, have significantly lower correlations with *iraP* or *crl* or each other. Despite this very strong correlation of insertions in *iraP* and crl during many growth conditions, during recovery from stress, they have dramatically different effects, with *crl* mutants less able to recover from stress (due to less RssB), while deletion of *iraP* made recovery more rapid, even after stabilizing conditions that are apparently independent of IraP.

## Discussion

### RpoS, stabilized during stress, is actively degraded when the stress ceases

RpoS is unstable, with a half-life of a few minutes or less, under non-stress conditions. Once a stress is encountered, proteolysis is blocked and RpoS accumulates. Stabilization of RpoS, combined with increased synthesis, rapidly leads to sufficient RpoS to turn on transcription of the RpoS regulon and activate the general stress response. Degradation depends on RpoS interaction with the adaptor protein RssB, which delivers RpoS to the ATP-dependent ClpXP protease. Given that ATP-dependent proteolysis is an energy-intensive process, there must be a

significant advantage to the cell regulating RpoS levels via proteolysis, rather than through other mechanisms. The work presented here, as well as other studies, strongly suggests that a major advantage is the ability of the cell to quickly and irreversibly reduce RpoS levels when the stress has passed [25,29]. Here, we focus on the ability to rapidly degrade RpoS after the stress is no longer present.

Previous work by others demonstrated that RpoS was stabilized during carbon starvation and that this stabilization was rapidly reversed when glucose was restored [8,29]. Here we find that restoration of RpoS proteolysis is rapid and robust once cells were released from three stabilizing stress conditions–phosphate starvation, carbon starvation, and stationary phase (Fig 2C and 2D). Stabilization is achieved in a different manner in each of these cases. During phosphate starvation, the anti-adaptor IraP is induced and is needed to stabilize RpoS. No anti-adaptor has been implicated in RpoS stabilization during carbon starvation. Instead, the lower levels of ATP that are available under this condition have been implicated in interfering with the ability of ClpXP to degrade of RpoS [29]. In stationary phase, at least two anti-adaptors, IraP and IraD, have been shown to be induced [30]. However, we find that even in a strain deleted for the genes for all three characterized anti-adaptors, *iraP*, *iraD* and *iraM*, RpoS was stabilized in stationary phase (Figs 2E and S1D). This variation in the mechanisms for stress stabilization make it particularly intriguing that the recovery phases show so many similarities. We suggest that the explanation for the similarities is that recovery from stress is controlled by Crl, and that this may be one of the more important roles of Crl.

## Requirements for an effective negative feedback loop: RpoS and Crl

We find that a negative feedback loop provides an essential component of recovery from stress (Fig 10). This was most clearly demonstrated using a degradable reporter fusion that behaves like RpoS–it is stabilized during phosphate starvation and is quickly degraded when phosphate is restored (Fig 3). However, when RpoS is not present, RpoS fusion levels are much higher (Fig 6A) and degradation is not restored after the stress ends (Fig 3B).

A critical component of this feedback loop is RpoS transcription of *rssB*, encoding the adaptor protein for RpoS degradation. *rssB* transcription is RpoS dependent; and artificially expressing higher levels of *rssB* bypassed the requirement for RpoS (Figs 4, 5C and S4) [24]. RssB levels increased gradually during phosphate starvation and were significantly lower in the absence of RpoS (Fig 4B). This RpoS-dependent regulation of RssB means that, during any stress in which RpoS accumulates, RssB levels will increase, and we find that this increase, while not promoting RpoS degradation during stress, is necessary for rapid recovery.

The operation of this feedback loop is sensitive to the presence of Crl, an activator of RpoS activity, stimulating its ability to successfully compete for core polymerase. Consistent with previous studies, mutations in Crl reduce the activity of a series of RpoS-dependent promoters, including *rssB* promoters (S4 Fig) [22,23]. Additionally, the absence of Crl leads to a significant increase in RpoS-mCherry fusions levels (Fig 6B). Finally, in the absence of active Crl, RpoS degradation during recovery from phosphate starvation and glucose starvation was significantly slowed (Fig 5), dependent on RpoS (Figs 5D, 5E, 6C and S7C). In the absence of Crl, less RpoS activity leads to less activity of the *rssB* promoters, less RssB, and thus more RpoS (Fig 6) [24]. Crl was previously found to be particularly important when RpoS levels are low [34]. Here we show that Crl serves as a critical regulatory component during recovery from stress, when RpoS levels are initially high. The gene coding for *crl* is interrupted by an IS element in many *E. coli* strains, including some variants of the reference K-12 strain MG1655 [41]. The loss of *crl* in many *E. coli* strains suggests that lower RpoS activity and decreased recovery capacity may present a fitness advantage in some environments [40,42].

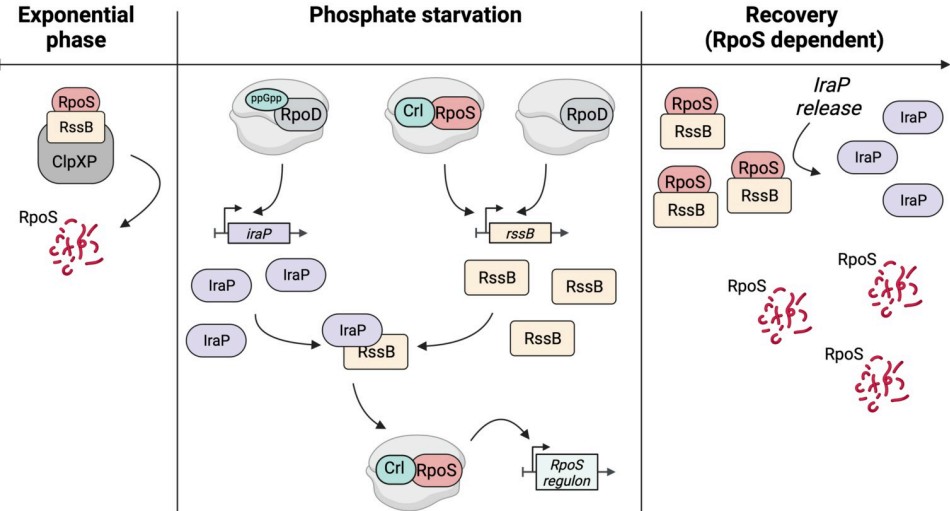

**Fig 10. Model for RpoS degradation after phosphate starvation.** This figure was created using clipart from BioRender.com.

At least one other feedback loop appears to exist, independent of RpoS degradation and of Crl, since a stable RpoS-mCherry fusion is still expressed at higher levels in the absence of RpoS (Fig 6D). It seems likely this second component affects the translation of RpoS, since the reporter fusions carry a synthetic promoter.

## The switch/release of RssB to degrade RpoS is not likely to rely on a change in IraP/RssB ratios

While RpoS-dependent transcription of *rssB* is necessary to allow degradation of RpoS during recovery, the increase in RssB levels per se does not appear to be a sufficient explanation for the resumption of RpoS degradation, since as long as starvation continues, RpoS remains stable. As the stabilization of RpoS during phosphate starvation is dependent on the ability of IraP to inhibit RssB, it was expected that the ratio of IraP to RssB would be a critical determinant of RpoS stability. *iraP* transcription is positively regulated by ppGpp, an alarmone that increases when cells are starved for various nutrients, including phosphate [10,32]. Once the stress has passed, ppGpp levels are likely to drop rapidly and new IraP will no longer be made. Here, we found that the tagged IraP protein levels increased after 30' of starvation, remained stable after the stress had passed and that the ratio of IraP to RssB does not significantly change (Fig 4C). IraP has not been observed to be degraded by ClpXP *in vitro* [9]. Thus, unlike sRNAs, there is no evidence that IraP is used stoichiometrically or is degraded when RpoS does not need to be stabilized anymore, but further work will be necessary to fully rule out degradation or inactivation of IraP during recovery.

We found that IraP but not IraD or IraM, affected recovery from stationary phase (Fig 2F). IraP, IraD and IraM are expressed under different conditions, are not structural homologs and each interact differently with RssB [9]. Whether IraD and IraM also allow rapid recovery after stresses under which they are expressed remains to be investigated.

A second attractive hypothesis was that RssB conformation might be altered through the phosphorylation of its receiver domain, as is the case for canonical response regulators. However, point mutants that either mimic the phosphorylated state of RssB or render RssB

unphosphorylatable did not impact its ability to degrade RpoS after phosphate starvation (Fig 7). Interestingly, under our growth conditions, the activity of a strain expressing the *rssB*-D58E (phosphomimic) or *rssB*+ were indistinguishable, suggesting that RssB was likely to be predominantly phosphorylated under these conditions.

## Our proposed model is still missing the basis of switch/release of RssB to degrade RpoS

We propose that RpoS recovery after phosphate starvation is linked to dissociation of the RssB-IraP complex, which occurs either by a change of conformation in RssB or IraP or by the involvement of some yet to be defined additional factor, leading to dissociation of IraP from RssB. Certainly, cell metabolism is changing dramatically as starvation ends, making it a challenge to identify the changes most important for restoring RpoS degradation. For instance, high ppGpp levels have global effects on transcriptional patterns, disfavoring core polymerase from RpoD in favor of RpoS. It has been demonstrated that the core RNA polymerase protects RpoS from degradation [24]. After phosphate is returned, a decrease in ppGpp would favor RpoD activity at those promoters, possibly leaving RpoS free from core polymerase and therefore more accessible to RssB for degradation. However, if that were a critical part of recovery, we would have expected a difference in the recovery kinetics of RpoS-LacZ, not expected to bind to core RNAP, and RpoS. This was not observed (Fig 3A). Consistent with competition for core not playing a central role in recovery, mutants in two RpoS-dependent factors that affect RpoD/core interactions, 6S RNA [38] and [39] had no significant effect on RpoS stabilization during starvation or RpoS degradation during recovery (Fig 8).

Another possible source of a switch, allowing RpoS degradation at the end of stress, could be due to changes in ATP status leading to an increased activity of the ClpXP protease to degrade RpoS. Stabilization of RpoS during glucose starvation has been attributed to limited ATP levels preferentially stabilizing RpoS by limiting the ability of ClpX to unfold RpoS and feed it to ClpP [29]. Certainly, as glucose is returned to cells, the ATP status should rapidly change. Is that sufficient to explain recovery? If so, it is not clear why the RpoS-dependent increase in RssB would be necessary for recovery, as it is even for glucose starvation. During phosphate starvation and recovery, it is clear that IraP is necessary for stabilization, suggesting that, in the absence of IraP, ClpXP is fully capable of efficiently degrading RpoS. If a change in ClpXP upon restoration of phosphate leads to recovery, that would suggest that ClpXP status helps push the equilibrium to RssB capture of RpoS rather than its sequestration by IraP.

Not explicitly discussed here but currently under study is whether IraP itself is the sensor of the switch, upon refeeding. A comparison of recovery from different stresses examined here hint at this possibility. While IraP is critical for stabilization during phosphate starvation, it has a minor role during glucose starvation and no detectable role in stationary phase (Figs 1E, 2D and S1). Nonetheless, recovery is very similar from all three stresses (Fig 2C), and, unexpectedly, IraP seems to play a role in providing some residual stabilization in all cases (Fig 2D, compare Δ*iraP* values for recovery). In addition, we note that IraP-SPA, while functional for stabilization, slows recovery, suggesting that characteristics of IraP (and its C-terminus) may be important for the recovery process. While this leaves the precise mechanism leading to RpoS recovery from these starvation conditions unknown, it suggests some fruitful directions for further research.

Overall, our results suggest that *E. coli* uses both IraP and Crl in parallel ways to modulate the RpoS regulon, with important and unexpected roles for both in not only adaptation to stress but also in the recovery from stress. These results highlight the precisely choreographed balance the cell maintains between the processes for regulating RpoS.

## Material and methods

### Media and growth conditions

All strains were grown either in LB (Lennox Broth, KD medical) or in MOPS minimal medium with shaking under aerobic conditions at 37˚C, unless otherwise indicated. The MOPS minimal medium was made with 10X MOPS medium buffer (Teknova) supplemented with 1.32 mM of potassium phosphate dibasic and 0.2% of glucose. When performing phosphate and glucose starvation experiments, MOPS medium was made similarly except that the potassium phosphate or glucose was omitted. Antibiotics were added to the medium to select and maintain plasmids and to select insertion mutations during strain construction as follows: 100 ug/mL of ampicillin, 50 ug/mL of kanamycin, 50 ug/mL of chloramphenicol, 25 ug/mL of tetracycline.

### Bacterial strains and plasmids

All bacterial strains, plasmids, primers, and DNA synthetic fragments used are listed in S1, S2, S3 and S4 Tables respectively. Note that one of the *rpoS*-dependent promoters used here is for gene *yodD*. *yodD* is transcribed divergently from *dsrB*, originally described as RpoS-dependent based on a transcriptional fusion containing the intergenic region between *dsrB* and *yodD* [36]. However subsequent studies of the transcription of *dsrB* in our lab did not confirm this, and it was found that the orientation of the promoter fragment in the original strain (DDS720) was in the opposite orientation, with the *yodD* promoter in fact driving the lacZ fusion. That RpoS-dependence of *yodD* is confirmed here.

All deletion or single amino acid substitution mutants were introduced into the correct background strains by P1 transduction as indicated in S1 Table. Transductants were selected on plates containing the appropriate antibiotic and clones were further purified on LB plates three times. Mutants containing antibiotic resistance markers were directly selected for that antibiotic, while non-marked mutants were moved into recipient strains carrying a nearby auxotrophic mutant marked with an antibiotic resistance insertion by co-transduction, selecting growth on minimal media and confirming loss of the antibiotic resistance marker.

### Cloning and mutagenesis

Primers used for cloning and mutagenesis are listed in S3 Table and gblocks used for cloning in S4 Table. For plasmid construction, Gibson assembly strategy (NEBiolabs) was used, and for site-directed mutagenesis, Quikchange mutagenesis (Stratagene) was used. Both methods were used according to the manufacturer's instructions. Plasmids are listed in S2 Table.

**Transcriptional fusions.** The pQE80L plasmid was used as the backbone for constructing plasmid-based transcriptional fusions of promoters of interest to mCherry, as shown in S4A Fig. First, the vector pQE80L-mCherry (pSB19) was constructed by inserting the mCherry coding sequence from the strain NM801 into pQE80L. Then, promoter sequences upstream of the genes *osmE* (from -201 to -12 bp from the ATG, creating pSB38), *osmY* (from -298 to -12 bp from the ATG, creating pSB22), *gadB* (from -160 to -12 bp from the ATG, creating pSB23) and *yodD* (from -162 to -12 bp from the ATG, creating pSB21), all followed by a RBS consensus sequence (AGGAGGTcagct), were inserted individually into pSB19 using Gibson assembly. The same *gadB*-mCherry fusion was inserted into the chromosome to create strain SB66, and its activity and RpoS-dependence were compared to that for the plasmid-borne fusion in pSB23 (S4E Fig); activities were very similar, as was the effect of deleting *rpoS*.

*rssB fusion measuring transcription and translation.* pSB37 is a derivative of pSB19 containing 1634bp upstream of the coding sequence of *rssB* and the first 24bp of the *rssB* gene, as

constructed in Pruteanu et al, 2002 (25), and includes both *rssB* promoters, P1 and P2 (see S3A and S4A Figs for details).

## Constructions of Δ*crl* and *crl*-R51A in MG1655 chromosome

To insert the deletion of *crl* and the R51A mutation of *crl* in the chromosome, λ red recombineering was used. Initially, a *Kan*-pBAD-*ccdB* cassette was introduced at the *crl* locus, in place of the coding region, to create strain SB98. *ccdB* encodes a toxin that inactivates DNA gyrase, and thus acts as a counter-selectable marker [43]. The cassette was amplified by PCR using the SB-24 and SB-39 primers, containing homology to *crl* flanking regions using chromosomal DNA from MG1655 containing an intact *crl* gene. The PCR product was purified, transformed, and recombined into the chromosome of strain NM1100, which contains genes for the bacteriophage λ red recombination system. The recombinants were selected at 32˚C on LB agar plates supplemented with 50 ug/mL of kanamycin and 1% glucose to repress the pBAD promoter. Clones were confirmed by PCR. The *kan*-pBAD-*ccdB* cassette in SB98 was then replaced by synthetic DNA fragments (gBlock, IDT) containing the flanking regions of *crl* and either the *crl* deletion or *crl* coding gene with R51A mutation (CGT 350–352 nucleotides into *crl* replaced by GCC) (see S4 Table for sequence of gblocks). Recombinants were selected at 37˚C on LB plates supplemented with 1% arabinose to induce the pBAD promoter and thus the expression of *ccdB*; only recombinants that have removed *ccdB* should survive. Clones were screened for loss of kanamycin resistance and confirmed by DNA sequencing. The *crl* markers were moved into other strains by first creating a proline-requiring strain, by moving a Δ*proB*::*kan* mutation into the strain, and then using this as a recipient, selecting for co-transduction on minimal glucose agar plates.

Deletion mutants were introduced into the correct background strains by P1 transduction as indicated in S1 Table. Transductants for antibiotic resistance markers were selected on plates containing the appropriate antibiotic and clones were further purified on LB plates three times.

## Constructions of *rssB*-D58 single mutants in MG1655 chromosome

To insert the D58A, D58P and D58E mutations of *rssB* in the chromosome, λ red recombineering was used. Initially, a *Kan*-pBAD-*kid* cassette was introduced at the *rssB* locus, in place of the coding region for amino acids 1 to 168 amino acids of *rssB*, to create strain AT485. *kid* encodes a toxin, and thus acts as a counter-selectable marker [44]. The cassette was amplified by PCR using the AT249 and AT254 primers, containing homology to *rssB (1–168 amino acids)* flanking regions using chromosomal DNA from MG1655. The PCR product was purified, transformed, and recombined into the chromosome of strain NM1100, which contains genes for the bacteriophage λ red recombination system. The recombinants were selected at 32˚C on LB agar plates supplemented with 50 ug/mL of kanamycin and 1% glucose to repress the pBAD promoter. Clones were confirmed by PCR. The *kan*-pBAD-*kid* cassette in AT485 was then replaced by synthetic DNA fragments (gBlock, IDT) containing the flanking regions of *rssB* and *rssB* coding gene with D58E, D58A or D58P mutations (see S4 Table for gblock sequences). Recombinants were selected on LB plates supplemented with 1% arabinose at 37˚C to induce the pBAD promoter and thus the expression of *kid*; only recombinants that have removed *kid* should survive. Clones were screened for loss of kanamycin resistance and confirmed by DNA sequencing. The tryptophan-requiring Δ*trpA*::*kan* mutant was introduced into strains by P1 transduction and those strains used as recipients to introduce *rssB* alleles, selecting for growth in the absence of trytophan.

## Constructions of Δ*rsd* in MG1655 chromosome

To delete *rsd* in the chromosome, λ red recombineering was used. Initially, a *Kan*-pBAD-*ccdB* cassette was introduced at the *rsd* locus, in place of the coding sequence, to create strain SB497. The cassette was amplified by PCR using the SB196 and SB197 primers, containing homology to *rsd* flanking regions using chromosomal DNA from MG1655. The PCR product was purified, transformed, and recombined into the chromosome of the strain NM1100, which contains genes for the bacteriophage λ red recombination system. The recombinants were selected at 32˚C on LB agar plates supplemented with 50 ug/mL of kanamycin and 1% glucose to repress the pBAD promoter. Clones were confirmed by PCR. The *kan*-pBAD-*ccdB* cassette in SB497 was then replaced by synthetic DNA fragments (gBlock, IDT) containing the flanking regions of *rsd* (see S4 Table for sequences). Recombinants were selected at 37˚C on LB plates supplemented with 1% arabinose to induce the pBAD promoter and thus the expression of *ccdB*. Clones were screened for loss of kanamycin resistance and confirmed by DNA sequencing.

## Construction of reporters P*gadB*-mCherry and *rpoS*-mCherry in the MG1655 chromosome

Fusions to mCherry were created starting with NRD1166. The NRD1166 strain carries the λ red functions at the λatt site and a *zeo-kan*-pBAD-*ccdB* counter-selectable marker preceding mCherry at the *lac* locus. To insert P*gadB*-mCherry in the chromosome, the *kan*-pBAD-*ccdB* cassette in NRD1166 was replaced by amplifying the *gadB* promoter region (from -160 to -12 bp from the ATG of *gadB*, as for constructing pSB23) by PCR using the primers SB-14 and SB-15. Recombinants were selected at 37˚C on LB plates supplemented with 1% arabinose to induce the pBAD promoter and the expression of *ccdB*. Clones were screened by PCR and confirmed by DNA sequencing. To create the RpoS-mCherry reporter, NRD1166 was electroporated with a PCR fragment made by amplifying the zeo-Cp17-*rpoS*750 fragment from BA754 genomic DNA, extending from the +1 of the *rpoS* promoter within the upstream *nlpD* gene through 750 nts of the ORF using primers RpoS750-mCherNRD1166F and RpoS750-mCherNRD1166R, containing the constitutive promoter Cp17. The PCR fragment also carries 40 nt homologies to the upstream *zeo* and downstream mCherry genes respectively. After electroporation, the cells were allowed to recover in LB- 1% glucose overnight on the bench and plated on LB-1% arabinose plates. The final insert was amplified and sequenced using primer set zeo-F and Int-mCh-sequencing primer and tested for fluorescent expression.

## Bacterial two-hybrid assay

An adenylate cyclase reconstitution-based bacterial two-hybrid (BACTH) system was used to assay protein interactions in vivo [45]. WT or R51A Crl and RpoS proteins were fused at the C-terminus of T25 and T18 fragments respectively which correspond to amino acids 1–224 and 225–399 of the CyaA protein, respectively. A derivative of *E. coli* strain BTH101 deleted for *rssB* and *rpoS* (SB71) was cotransformed with both plasmids and plated on LB-agar medium containing 100 ug/mL ampicillin and 50 ug/mL kanamycin and incubated at 32˚C for 48 hours. To quantify the interactions, cells were grown in 3 mL LB containing 100 ug/mL ampicillin, 50 ug/mL kanamycin and 0.5 mM IPTG at 32˚C overnight. Beta-galactosidase activities were calculated by measuring the kinetics of appearance of the ONPG degradation product at OD = 420 nm every 5 seconds for 20 minutes. The beta-galactosidase activity was calculated using the slope of $OD_{420}$ corrected with the OD600 of the cells.

## RpoS recovery after stress

*Phosphate and glucose starvation.* Cells were grown in MOPS minimal glucose medium from a starting $OD_{600}$ of 0.05, to an $OD_{600}$ of 0.3 at 37˚C. Cells were filtered, washed 3 times with MOPS minimal medium containing either no phosphate or no glucose and were grown in this medium for 1 hour at 37˚C. A one-mL aliquot was taken, and phosphate or glucose was subsequently added back to the remaining cultures. One-mL aliquots were taken at different time points. In the case of chase experiments, chloramphenicol was added 2 min after phosphate or glucose was added back. The one mL aliquots were then treated by TCA precipitation as follows. The aliquots were directly added into 110uL of cold TCA and kept on ice for at least 30 min. After centrifugation for 10 min at 13K at 4˚C, the supernatant was discarded and 500uL of cold acetone was added. After a second centrifugation step, the supernatant was removed and the pellets were left for complete drying overnight on the bench. Loading buffer was then added, normalizing so that 50uL correspond to $OD_{600} \approx 0.3$.

*Stationary phase recovery.* Cells were grown in MOPS minimal glucose medium overnight at 37˚C. Cultures were then diluted 5-fold into fresh MOPS minimal glucose medium and chloramphenicol was added 2 minutes after to start the chase. One-mL aliquots were taken at different time points; the $t_0$ time point was taken before dilution.

## Western blot analysis

TCA-treated samples were loaded onto Nu-PAGE 12% Bis-Tris gel (Invitrogen). The proteins were transferred to nitrocellulose membrane using iBlot Gel transfer block (Novex Life Technologies), blocked using Blocking Buffer (Bio-Rad), probed with polyclonal rabbit anti-RpoS (1:5000 dilution), mouse anti-Flag (Sigma, 1:2000 dilution), mouse monoclonal anti-EF-TU (LSBio, LS-C128699) (1:10,000), rabbit polyclonal anti-RssB (1:5000 dilution), and further probed with the secondary antibodies, either goat anti-mouse coupled with DyLight 800 (Bio-Rad, 1:10000 dilution) or goat anti-rabbit coupled with StarBright Blue 700 (Bio-rad, dilution 1:5000) and washed with PBST solution. The blots were visualized using fluorescent exposure with the ChemiDoc imager (Bio-Rad). Quantification was performed using Image J software (NIH).

## Measurement of the activity of transcriptional fusions

Cells were grown overnight in LB at 37˚C, with ampicillin at 50ug/mL for strains containing the transcriptional fusions on pQE80L vectors to maintain the plasmid. Cultures were then diluted at 1/100 in 1 ml MOPS minimal medium supplemented with ampicillin 50ug/mL when needed in a 24-well microplate. Expression of the transcriptional fusion was measured by growing cells in a microplate reader (Tecan) that reads both mCherry fluorescence (excitation at 587 nm and emission at 610 nm) and $OD_{600}$ every 20 minutes for 18 hours at 37˚C with shaking. Fluorescence values were divided by the $OD_{600}$ values to obtain relative fluorescence units (RFU).

## Supporting information

**S1 Fig. IraP participates in RpoS recovery after glucose starvation and during stationary phase.** A) Western blots showing RpoS accumulation (no chloramphenicol) after phosphate was added to the MG1655 strain (recovery) or when incubation was continued without phosphate addition. Results are plotted in Fig 1D. B) Western blot against RpoS and the loading control EF-Tu showing RpoS degradation (chase) during (stabilization) and after exit from stationary phase (recovery) in MG1655 and Δ*iraP* (SB151) strains. Cells were grown in MOPS

minimal glucose medium overnight to reach stationary phase. Stationary-phase cells were diluted 5-fold back into fresh medium and chloramphenicol was added after 2 minutes. Samples were taken and treated as described for Fig 1. The graph on the right corresponds to quantification of RpoS degradation (n > 3). C) Western blot against RpoS and the loading control EF-Tu showing RpoS degradation (chase) during and after glucose starvation in MG1655 and ΔiraP (SB151) strains. Samples were taken and treated as described for Fig 1. The graphs on the right and at the bottom correspond to quantification of RpoS degradation (n > 3). D. Stationary Phase, RpoS chase in ira mutants. Western blot of RpoS and the loading control EF-Tu showing RpoS degradation after Chloramphenicol addition (chase) during (Stab, for stabilization) and after exit from stationary phase (Rec for recovery) as in B, but in the following isogenic derivatives of MG1655, carrying different combinations of ira mutant alleles. Strains used: ΔiraP (SB151); ΔiraD (SB364); ΔiraM (SB539); ΔiraP ΔiraD (SB365); ΔiraP ΔiraM (SB540); ΔiraD ΔiraP (SB541); ΔiraD ΔiraM ΔiraP (SB542). Quantitation of triplicates is shown in Fig 2D (stabilization) and 2E (recovery).
(PDF)

**S2 Fig. RpoS-Lac recovery from phosphate starvation depends on RpoS.** A) The RpoS-Lac fusion protein is inactive for transcription. The plasmid pSB23 bearing a transcriptional fusion between the RpoS-dependent promoter PgadB and mCherry (full description of the fusions and method of measurement described in S4 Fig) was introduced in strains containing RpoS-Lac in the presence (SG30013) and absence of RpoS (INH28). The loss of the mCherry signal upon deletion of rpoS confirms that the fusion protein is not able to activate an RpoS-dependent promoter. B) Western blot showing RpoS and RpoS-Lac stabilization and degradation during phosphate starvation and recovery in the strains containing the RpoS-Lac translational fusion in the presence of RpoS (strain SG30013), and in the absence of RpoS (INH28). The protocol is as in Fig 1, with chloramphenicol added to stop translation. C) Quantitation of RpoS and RpoS-Lac half-life during phosphate starvation from Western Blot as shown in S2B Fig (n = 3). (D) Quantitation of RpoS and RpoS-Lac half-life during recovery from phosphate starvation, from Western Blot as shown in S2B Fig (n = 3).
(PDF)

**S3 Fig. *rssB* transcription and RssB protein induction during phosphate starvation and recovery.** A) P1 and P2 rssB promoter sequences and fusion schematics. 1597pb upstream of the rssB coding gene and the first 24 bp of the rssB gene were fused to mCherry. The start point for the P1 promoter is located 1135pb upstream of the first rssB codon and 139 nt upstream of the first rssA codon; P2 is located 64 bp upstream of the first rssB gene codon. Note that the position of the P2 start was determined from dRNA-seq data determined by Thomason et al [46] and differs from the start site previously identified for P2. B) Western Blot of RssB levels during phosphate starvation and after phosphate was added back (phosphate recovery) in MG1655 (WT). C) Western Blot of RssB levels during phosphate starvation (1–60' samples) and one hour after phosphate was added back (120' sample) in WT and ΔrpoS (AB165) strains. D) IraP-SPA levels during glucose starvation, phosphate starvation (0, 10, 30 and 60-minute samples for each), exponential phase ($OD_{600}$ = 0.5) and stationary phase ($OD_{600}$ = 1.4). Western blot against FLAG-tag of samples from a strain in which a SPA tag was inserted at the C-terminus of IraP in the chromosome (strain SB212) and against the loading control GroEL. Phosphate and glucose starvation follow the protocol as described in Figs 1 and 2. Note that we were unable to detect untagged IraP by western blot. E) Stability of IraP-SPA protein during and after phosphate starvation. Chase experiment in the strain SB212 containing iraP-SPA at the iraP locus and Western Blot against Flag-tag detecting the SPA tag of IraP. Samples were taken from cells undergoing phosphate starvation; chloramphenicol was

added to start the chase as described in Figs 1 and 2. F) Effect of IraP-SPA tag on RpoS stabilization and recovery. RpoS chase during and after phosphate starvation in strains WT (MG1655) and SB212 containing *iraP*-SPA at the *iraP* locus, following the protocol as described in Fig 1A. G) RpoS chase during phosphate starvation in strains WT (MG1655) and SB212 containing *iraP*-SPA at the *iraP* locus, following the protocol as described in Fig 1A. (PDF)

**S4 Fig. Crl activates the transcription of many RpoS-dependent promoters including the *rssB* promoter.** A) Plasmid maps for construction of transcriptional fusions. The empty vector pQE80L was used as the backbone to construct the pSB19 plasmid by Gibson assembly, replacing the Lac promoter/operator and the RBS by the coding sequence of mCherry following a consensus RBS sequence (see dotted lines for both pQE80L and pSB19). The promoter regions of *yodD*, *osmY*, *gadB* and *osmE* or the 1658 bp upstream of *rssB* were then introduced into pSB19, obtaining the plasmids pSB21, pSB22, pSB23, pSB38 and pSB37 respectively. B) The activity of the transcriptional mCherry fusions of the RpoS-dependent promoters of *yodD* (pSB21, top left panel), *osmY* (pSB22, top right panel), *gadB* (pSB23, bottom left panel) and *osmE* (pSB38, bottom right panel) in the WT, Δ*rpoS* (AB165), Δ*crl* (INH24), *iraP*::kan (AB006) and *iraP*::kan Δ*crl* (INH26) strains. Strains were grown from exponential to stationary phase in a microplate reader in MOPS minimal medium that measured mCherry fluorescence and OD600 every 20 minutes. The transcriptional fusion construct is shown above each graph (see Material and methods for details). C) Comparison of relative average expression of the transcriptional fusions of the four RpoS-dependent promoters to WT, after 16 hours of growth, from data as in S4B (n > 3). D) Relative expression of the four RpoS-dependent transcriptional fusions in the Δ*rpoS* (AB165), Δ*crl* (SB147), Δ*iraP* (SB151) and Δ*iraP* Δ*crl* (INH26) strains, with WT (MG1655) set to 100, during stationary phase. E) Comparison of the activity of the *gadB* promoter fused to mCherry, expressed either from the pQE80L-derived plasmid pSB23 or from the same fusion expressed as a single chromosomal copy, in WT (MG1655; blue symbols) and Δ*rpoS* (AB165; red symbols) strains. Strains were grown and the RFU measured as described in Fig 4A. F) Fluorescence over time of the translational fusion contained on pSB37, carrying the upstream region of *rssB* fused to mCherry, as shown in S4 Fig. The fusion extends 1597bp upstream of the *rssB* coding gene, including the P1 and P2 promoters, as shown in S3A and S4A Figs. The strains used in S4B Fig were transformed with the plasmid containing the *rssB* fusion and followed during growth. (PDF)

**S5 Fig. Rapid RpoS recovery from phosphate and from glucose starvation requires Crl.** A) Western blot against RpoS showing RpoS degradation during recovery from phosphate starvation in the strains WT, *crl*-R51A (SB148) and Δ*crl* (SB147), primary data for Fig 5A. Experiment was performed as described in Fig 1A. B) Western blot against RpoS showing RpoS degradation after glucose starvation in the WT and Δ*crl* (SB147) strains, primary data for Fig 5B. Experiment was performed as described in Fig 1. C) Western blot and quantification of RpoS degradation during phosphate starvation in WT, *crl*-R51A (SB148) and Δ*crl* (SB147) strains. Experiment was performed as described in Fig 1A. D) Western blot and quantification of RpoS degradation during glucose starvation in WT and Δ*crl* (SB147) strains. Experiment was performed as described in Fig 1A. E) RpoS half-lives in MG1655 and Δ*crl* (SB147) strains during (stress) and after (recovery) phosphate starvation and glucose starvation as determined in experiments shown in Figs 5A, 5B, S5A, S5B, S5C and S5D. RpoS half-lives correspond to the time at which half of the $t_0$ RpoS protein disappears in a chase assay. ND: Not determined. (PDF)

**S6 Fig. Crl-dependent RpoS feedback loop.** A) Bacterial two-hybrid experiment showing the interaction between RpoS and Crl. A plasmid expressing the T25 domain of adenylate cyclase was fused to the *rpoS* coding gene at its 5' end and a plasmid expressing the T18 domain was fused to the wild-type *crl* or *crl*-R51A coding gene at its 5' end. Western Blot detecting the T18 domain of the adenylate cyclase (below graph) shows similar production of T18-Crl and T18-Crl-R51A in the *cya*+ MG1655 strain. B) Fluorescence over time of the translational *rssB* fusion between mCherry contained on the pSB37 plasmid in WT, Δ*rpoS* (AB165), Δ*rssB* (SB94), Δ*crl* (SB147) and *crl*-R51A (SB148) strains, grown from exponential to stationary phase in a microplate reader that measured mCherry fluorescence and OD600 every 20 minutes. C) Western blot against RpoS showing RpoS and RpoS-Lac recovery accumulation from phosphate starvation in WT, *iraP*::*kan* and *crl*::*kan* strains. *rpoS*+ strains: SG300013 (*crl*+ *iraP*+), SB179 (*iraP*-) and SB180 (*crl*-). Δ*rpoS* strains: INH28 (*crl*+), SB175 (*iraP*-) and SB176 (*crl*-). Δ*rpoS rssA2*::*cm* strains: SB150 (*crl*+), SB173 (*iraP*-) and SB174 (*crl*-). D) Western blot showing RpoS and RpoS-Lac accumulation during phosphate starvation in the *crl*+ *iraP*+ strains used in C. Protocol is as in Fig 1. E) Relative RpoS-Lac levels from the Western Blot in S6D Fig at 0' time points (pre-phosphate starvation). Values are represented as percentage relative to RpoS-Lac levels in the *rpoS*+ strain, set to 100.
(PDF)

**S7 Fig. Crl effect on RpoS levels depends on RpoS and RssB.** A) The fluorescence of the degradable RpoS-mCherry fusion in WT (NM801), Δ*rpoS* (SB238), *rssA2*::*cm* (SB226), Δ*crl* (SB228), Δ*rpoS rssA2*::*cm* (SB281), Δ*rpoS* Δ*crl rssA2*::*cm* (SB280) and Δ*crl rssA2*::*cm* (SB282) strains was measured as described in Fig 4A. B) The fluorescence of the degradable RpoS-mCherry fusion in WT (NM801), Δ*rpoS* (SB238), Δ*rpoS* Δ*crl* (SB239), Δ*crl* (SB228), Δ*rssB* Δ*crl* (SB283), and Δ*rssB* (SB225) strains was measured as described in Fig 4A. C) Ratios between RFU values in strains *crl*- / *crl*+ in WT, *rpoS*- and *rssB*- strains, from the data shown in S7B Fig. WT is set to 100. D) Full graphs of the ratios between the strains deleted of *crl* (Δ*crl* (SB228), Δ*crl* Δ*rpoS rssA2*::*cm* (SB280) and Δ*crl rssA2*::*cm* (SB282)) and the corresponding *crl*+ strains (WT (NM801), Δ*rpoS rssA2*::*cm* (SB281) and *rssA2*::*cm* (SB226)). Fluorescence and data analysis were performed as described in Fig 4A, with data during growth from S7A Fig. WT is set to 100. E) The fluorescence of the non-degradable RpoS-mCherry fusion in WT (NM802), Δ*rpoS* (SB439) and Δ*crl* (SB437) strains was measured as described in Fig 4A. F) The fluorescence of the chromosomally-inserted P*gadB*-mCherry transcriptional fusion from WT (SB66), Δ*rpoS* (SB67), Δ*rssB* (SB242), Δ*crl* (SB243), Δ*rpoS* Δ*rssB* (SB298), Δ*rssB* Δ*crl* (SB297) and Δ*crl* Δ*rpoS* (SB296) strains was measured as described in Fig 4A. G) The effect of the absence of Crl on P*gadB*-mCherry. mCherry fluorescence of the strains WT (SB66), Δ*rssB* (SB242), Δ*crl* (SB243) and Δ*rssB* Δ*crl* (SB297), all containing the chromosomally-inserted P*gadB*-mCherry fusion was measured over time as described in Fig 4A. Using the data from *rpoS*+ strains in S7F Fig, the RFU ratios during growth between the strains containing deletion of *crl* compared to *crl*+, in WT or Δ*rssB* strains was calculated, with the value in the *crl*+ strain across growth set to 100. The arrows indicate the ratios at times 200 and 800 minutes.
(PDF)

**S8 Fig. The phosphorylation status of RssB does not impact RpoS recovery from phosphate starvation.** Western Blots against RpoS and the loading control EF-Tu showing RpoS degradation after phosphate starvation of the strains WT, *rssB*-D58E (SB192), *rssB*-D58A (SB190) and *rssB*-D58P (SB198). Primary data for Fig 7B.
(PDF)

**S9 Fig. Impact of Rsd and 6SRNA on RpoS degradation after phosphate starvation.** Primary data for Fig 8. A) Western blot against RpoS and the loading control EF-Tu showing RpoS stabilization during phosphate starvation in MG1655, Δ*iraP* (SB151), Δ*ssrS::kan* (SB470) and Δ*rsd* (SB505) strains. Samples were taken and treated as described in Fig 1A, chloramphenicol was added to stop translation. B) Western blot against RpoS and the loading control EF-Tu showing RpoS degradation during recovery from phosphate starvation in MG1655, Δ*iraP* (SB151), Δ*ssrS::kan* (SB470) and Δ*rsd* (SB505) strains. Samples were taken and treated as described in Fig 1A, chloramphenicol was added to stop translation.
(PDF)

**S1 Table. Strains used in this study.**
(PDF)

**S2 Table. Plasmids used in this study.**
(PDF)

**S3 Table. Primers used in this study.**
(PDF)

**S4 Table. gBlock gene fragments used in this study.**
(PDF)

**S1 Data. Supporting original data files for all Figures.**
(ZIP)

## Acknowledgments

We thank members of the Gottesman and Wickner labs for discussion and advice throughout this work, and thank Aurelia Battesti, Michael Maurizi and Taran Bauer for comments on the manuscript.

## Author Contributions

**Conceptualization:** Sophie Bouillet, Issam Hamdallah, Arti Tripathi, Susan Gottesman.

**Formal analysis:** Sophie Bouillet, Susan Gottesman.

**Investigation:** Sophie Bouillet, Issam Hamdallah, Nadim Majdalani, Arti Tripathi.

**Methodology:** Sophie Bouillet, Nadim Majdalani.

**Supervision:** Nadim Majdalani, Susan Gottesman.

**Writing – original draft:** Sophie Bouillet, Susan Gottesman.

**Writing – review & editing:** Sophie Bouillet, Arti Tripathi, Susan Gottesman.

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
