## [Decision Letter · Decision Letter 0]

12 Dec 2023

Dear Dr Gottesman,

Thank you very much for submitting your Research Article entitled 'A negative feedback loop is critical for recovery of RpoS after stress in Escherichia coli.' to PLOS Genetics.

The manuscript was fully evaluated at the editorial level and by independent peer reviewers. The reviewers appreciated the attention to an important topic but identified some concerns that we ask you address in a revised manuscript.

We therefore ask you to modify the manuscript according to the review recommendations. Your revisions should address the specific points made by each reviewer.

Yours sincerely,

Kai Papenfort

Academic Editor

PLOS Genetics

Lotte Søgaard-Andersen

Section Editor

PLOS Genetics

Dear Dr Gottesman.

Thank you again for submitting your manuscript to PLOS Genetics. Your manuscript has now been evaluated by three referees. As you will see from the comments below, all three referees are enthusiastic about your

work and request only few changes to the manuscript. Please make sure to respond to all the comments in your revised manuscript and the rebuttal letter.

Best regards.

Kai Papenfort

Academic Editor

Reviewer's Responses to Questions

**Comments to the Authors:**

Reviewer #1: The general stress sigma factor S (RpoS) of E.coli is an important master transcription factor at the center of a complex regulatory network orchestrating different stress responses, in addition to biofilm formation, motility and last but not least the transition into stationary phase. RpoS expression, activity and stability is controlled on many different levels, including regulatory proteolysis.

Here RpoS is targeted by an adaptor protein RssB for degradation by the ClpXP AAA+ protease complex. RssB is a response regulator whose expression and also activity can be controlled. RssB can interact with RpoS inducing a conformational change, which exposes a degron, targeting it for ClpXP degradation. Interestingly a number of so-called anti-adaptor proteins were identified, whose expression is induced by different stresses (e.g. Phosphate, Mg, DNA repair) and once synthesized they can interfere with the RssB activity, e.g. by sequestration, which results in a stress signal dependent stabilization and activation of RpoS.

In their manuscript "A negative feedback loop is critical for recovery of RpoS after stress in Escherichia coli" the authors observed that a deletion of the gene for anti-adaptor protein IraP (which is induced during phosphate starvation) plays a RpoS stabilizing role not only during the stress response but to a certain extent also during recovery from phosphate starvation stress response. In addition, the absence of IraP influences RpoS stability during escape from the other two tested stress conditions (glucose starvation or stationary phase) (Fig 2D).

Further experiments demonstrated that the recovery from phosphate starvation also depends on the negative feedback loop of RpoS dependent transcription of rssB and the presence of Crl which stabilizes the RnaPol/RpoS Holoenzyme, known to further facilitate this negative feedback loop. Both proteins, IraP and Crl, appear to support RpoS activity and stability in parallel processes.

General comments

-This is a very interesting and comprehensive manuscript entailing many well-designed experiments to further characterize, examine and investigate the molecular mechanisms controlling the escape from stress response pathways. This is interesting since thereby it can be investigated what happens when the conditions for a bacterial population are improving and the cells can for example restart growth and resume to replicate and divide again.

-It is interesting to note that the three tested more complex stress conditions all could include for example the synthesis of the (p)ppGpp alarmone as part of the stress response. Maybe the common regulatory effect detected for IraP during recovery from stress responses, might be caused by such a commonality in the stress responses, and in the case of (p)ppGpp, might also coordinate with the transcriptional and also translational restart upon lowered alarmone levels.

-It could be discussed how the interaction of IraP and possibly the two other anti-adaptor proteins with RssB help to integrate the different signals and how IraP and the other adaptor proteins interact with RssB to control this proteolytic switch, especially when it has to be reverted.

-Especially for the reasons given in line 241-247 I believe that the stability (degradation) of IraP should not be totally ruled out as a mechanism to explain its role in escape from the stress conditions.

Minor comments

Somehow the references appear as if they could have been misplaced maybe upon reformatting. It would be good to systematically check them.

For example:

-Ref (28) (Pruteanu et al) in line 152 and line 460 should probably be (27) (Peterson et al)?

-Ref (29,33) in line 201 should probably be (28) (Pruteanu et al)?

Ref (25) in line 253 should probably be rather (24 (Typas et al) and maybe other references on the role of Crl in the feedback loop?

Ref (30,31) in line 460 Ref 31 (Becker et al 1999) describe the regulated proteolysis of RpoS via RssB and discuss the Bacillus competence regulatory proteolysis (including ComS which is also a kind of anti-adaptor protein) but at that time the Ira anti-adaptor proteins were not yet known.

Ref 45 in line 518 does not describe the website mentioned and described in this paragraph. The correct reference seems to be cited in the figure legend.

A typo in line 163

Reviewer #2: Numerous studies have investigated the induction of bacterial stress responses, yet the mechanisms governing bacterial recovery from stress are poorly understood. The authors asked how E. coli reverses the stabilization of the sigma factor RpoS in the recovery phase. They focused on two proteins, the anti-adaptor IraP and Crl, an activator of the RpoS-RNA polymerase holoenzyme. These proteins collectively contribute to a negative feedback loop that restores basal RpoS levels when the stress is gone. Despite these insights, the exact mechanism remains unsolved.

This is a solid study that will be of interest to microbiologists working on stress responses. The experimental design is clever, and the results support the conclusions. While the manuscript is carefully prepared and well written, there may be room for streamlining in various sections. For someone unfamiliar with the field, the information is very dense and rich in details. The Discussion spanning over more than eight pages appears somewhat lengthy. Trimming of detailed recapitulations of experimental results, for example on Crl mutants (page 24) or RssB phosphorylation (page 27), could enhance conciseness. Otherwise, I only have a few relatively minor comments.

I understand why the study focusses on IraP. Given that there are some gaps in the overall understanding, I was wondering whether the authors tested if and how the other two RssB-binding anti-adaptors, IraD and IraM, contribute to the recovery process.

Line 585: Please correct ItaP to IraP.

Reviewer #3: This paper is an important contribution to our understanding of bacterial stress responses. To date, the vast majority of work on bacterial stress responses focuses on how those responses are induced by a stress. This work focuses on the other half of that coin- how the response is turned off once a stressful condition ends. This is an important conceptual framing.

This paper focuses on the RpoS protein of E. coli, a well studied alternative sigma factor that accumulates in response to stress. Regulation of RpoS levels occurs at multiple levels, with regulated proteolysis playing an important role. Degradation of RpoS by the ClpXP protease requires the adaptor protein RssB. RssB's ability to interact with RpoS is regulated by small proteins called anti-adaptors, which were discovered in the senior author's lab. This paper demonstrates that active degradation of RpoS restarts upon relief from the stress, and that this restart involves a feedback loop with RssB as well as the regulator Crl. The authors are not able to determine the molecular mechanism kicking off this active degradation cycle, but they are able to eliminate some likely possibilities.

I have no concerns with the design or execution of the experiments. My only concerns with data interpretation have to do with a few places where it is difficult for me to assess the authors' interpretation of the data:

Line 245: I cannot see data in Figure 4E to support the claim that the SPA tag interfered with rapid recovery. In fact, I can't really see data for the iraP-SPA blots at all. I assume that they are hiding under the wild-type data, but I can't see which set of wild-type data.

Line 290 - 293 / Figure S5: The authors interpret the data in Figure S5 to show that crl is important for recovery of degradation but that crl doesn't play a role during starvation itself. In the plots that are part of Figures S5C/D, it looks like crl mutants do have lower half-lives than wild-type. Looking at the blots, the effect of crl looks to be larger during recovery, but it is difficult for me to make that assessment based only on the blots. This argument would be stronger if there were graphs for parts A and B, and then the authors computed the half-lives for the various genotype/condition combinations.

Beyond that, I have a few comments about the writing that I think will make the paper easier to read:

Line 114 and many other subsequent places: I found the "chase" designation very confusing because I believe the authors are using it in a nonstandard way. Protein turnover can be determined from a pulse-chase experiment, which the Oxford Dictionary of Biochemistry and Molecular Biology (2 ed.) describes as:

"a technique whereby cells growing in culture are exposed for a short period (pulse) to a radiolabelled molecule such as an amino acid and then transferred to a medium containing an excess of the nonlabelled substance for a longer period (chase). By removing samples at intervals the fate of intracellular components labelled during the pulse can be ascertained."

The experiments in this paper have no pulse of labeled molecules, so there is no chase with non-labeled molecules. Instead, all protein synthesis is stopped with the addition of an antibiotic, so the change in protein levels are due solely to degradation with no contribution of protein synthesis. I would suggest that the authors find some different nomenclature to distinguish between experiments that measure the net accumulation minus degradation vs. those that measure degradation only. Other studies from this lab that simply mark data as "minutes after Cm addition" were clear to me and I wonder if something describing which samples had chloramphenicol vs. which did not would be more straightforward.

Line 147-149: This statement is unclear to me. Are you saying that IraD plays a minor, not major role? Or are you saying that IraD plays a major role in the transition to stationary phase but a minor role during stationary phase? Or something else?

Line 163: "bes" is a typo

Lines 424 - 447: This reads more like introduction than discussion. I think the discussion would be improved by immediately letting the reader know major experimental findings.

Lines 514 - 526: I love the idea of using these data. A new analysis using existing data is still a new result, so this section should be in the results section, rather than in the discussion.

**Have all data underlying the figures and results presented in the manuscript been provided?**

Reviewer #1: Yes

Reviewer #2: Yes

Reviewer #3: Yes

PLOS authors have the option to publish the peer review history of their article (what does this mean?). If published, this will include your full peer review and any attached files.

Reviewer #1: No

Reviewer #2: No

Reviewer #3: No

---

## [Decision Letter · Decision Letter 1]

17 Jan 2024

Dear Dr Gottesman,

We are pleased to inform you that your manuscript entitled "A negative feedback loop is critical for recovery of RpoS after stress in Escherichia coli." has been editorially accepted for publication in PLOS Genetics. Congratulations!

Yours sincerely,

Kai Papenfort

Academic Editor

PLOS Genetics

Lotte Søgaard-Andersen

Section Editor

PLOS Genetics

Comments from the reviewers (if applicable):

Reviewer's Responses to Questions

**Comments to the Authors:**

Reviewer #1: I am happy with the revision

Reviewer #2: I am satisfied with the revision.

Reviewer #3: Thank for dealing thoughtfully with all of my concerns.

**Have all data underlying the figures and results presented in the manuscript been provided?**

Reviewer #1: None

Reviewer #2: Yes

Reviewer #3: None

PLOS authors have the option to publish the peer review history of their article (what does this mean?). If published, this will include your full peer review and any attached files.

Reviewer #1: No

Reviewer #2: No

Reviewer #3: No

**Data Deposition**

http://datadryad.org/submit?journalID=pgenetics&manu=PGENETICS-D-23-01259R1

**Press Queries**

---

## [Editor Report · Acceptance letter]

6 Mar 2024

PGENETICS-D-23-01259R1 

A negative feedback loop is critical for recovery of RpoS after stress in Escherichia coli. 

Dear Dr Gottesman, 

We are pleased to inform you that your manuscript entitled "A negative feedback loop is critical for recovery of RpoS after stress in Escherichia coli." has been formally accepted for publication in PLOS Genetics! Your manuscript is now with our production department and you will be notified of the publication date in due course.

With kind regards,

Anita Estes

PLOS Genetics

On behalf of:
